# Downstream rounding rate of pebbles in the Himalaya

Prakash Pokhrel[1,2], Mikael Attal[1], Hugh D. Sinclair[1], Simon M. Mudd[1], and Mark Naylor[1]

[1]School of GeoSciences, University of Edinburgh, UK
[2]Department of Mines and Geology, Nepal

**Correspondence:** Prakash Pokhrel (pokhrel.prakash@ed.ac.uk)

**Abstract.** Sediment grains are progressively rounded during their transport down a river. For more than a century, Earth scientists have used the roundness of pebbles within modern sediment, and of clasts within conglomerates, as a key metric to constrain the sediment's transport history and source area(s). However, the current practices of assessment of pebble roundness are mainly qualitative and based on time consuming manual measurement methods. This qualitative judgement provides the transport history only in a broad sense, such as classifying distance as 'near' or 'far'. In this study, we propose a new model that quantifies the relationship between roundness and the transport distance. We demonstrate that this model can be applied to the clasts of multiple lithologies including modern sediment as well as conglomerates deposited by ancient river systems. We present field data from two Himalayan catchments in Nepal. We use the Normalized Isoperimetric Ratio ($IR_n$) which relates a pebble's area ($A$) to its perimeter ($P$), to quantify roundness. The maximum analytical value for $IR_n$ is 1, and $IR_n$ is expected to increase with transport distance. We propose a non-linear roundness model based on our field data, whereby the difference between a grain's $IR_n$ and the maximum value of 1 decays exponentially with transport distance, mirroring Sternberg's model of mass loss or size reduction by abrasion. This roundness model predicts an asymptotic behaviour for $IR_n$, and the distance over which $IR_n$ approaches the asymptote is controlled by a rounding coefficient. Our field data suggest that the roundness coefficient for granite pebbles is nine times that of quartzite pebbles. Using this model, we reconstruct the transport history of a Pliocene paleo-river deposit preserved at the base of the Kathmandu intermontane Basin. These results, along with other sedimentary evidence, imply that the paleo-river was much longer than the length of the Kathmandu Basin, and that it must have lost its headwaters through drainage capture. We further explore the extreme rounding of clasts from Miocene conglomerate of the Siwaliks Zone and find evidence of sediment recycling.

## 1 Introduction

The rounding of pebbles found within conglomerates has long been linked to abrasion that occurs prior to deposition as pebbles are transported by rivers, with greater rounding being typically associated with increased transport distance (Mills, 1979; Russell, 1980; Lindsey et al., 2007; Yingst et al., 2016). This also applies to modern rivers where the shape of pebbles has been used to locate sediment sources and define the controls exerted by hydraulic transport on abrasion processes (Wentworth, 1919; Lindsey et al., 2005; Domokos et al., 2009; Litty and Schlunegger, 2017; Gale, 2021). The use of pebble roundness is not limited to Earth; research on Mars has connected roundness to both the existence of ancient river networks as well as the transport history of Martian sediments (Yingst et al., 2008; Jerolmack, 2013; Williams et al., 2013; Szabo et al., 2015). Several

researchers have attempted to relate pebble roundness with sediment transport distance based on field measurements (Wentworth, 1922; Mills, 1979; Roussillon et al., 2009; Litty and Schlunegger, 2017) as well as laboratory experiments (Wentworth, 1919; Abbott and Peterson, 1978). However, the method of assessing roundness is still mainly qualitative; as such, there is no model that quantitatively relates roundness over a long transport distance. We aim to fill this gap in this work. We first present an overview of our understanding of pebble shape changes through abrasion, shape indices and measurement techniques, before providing more details about the motivation for this work.

## 1.1 Pebble abrasion and shape change

The morphology of sediment grains plucked from bedrock, sourced from hillslopes and/or re-worked from existing deposits gets modified as the grains are transported downstream (McPherson, 1971). It is known that shape, size, and roundness evolve mainly due to abrasion processes that have acted upon the grain in time and space (Brewer and Lewin, 1993). Since the terminology used in the published literature may vary, we clarify that we use the term 'abrasion process' to broadly describe processes that lead to mass loss of grains due to energetic impact during fluvial transport (similar to what Miller and Jerolmack (2021) describe as 'attrition'). These processes include the small-scale breaking off of edges (chipping), corners and other fragments due to impacts during fluvial transport (e.g., (Miller et al., 2014; Szabo et al., 2015; Novák-Szabó et al., 2018)). We use the term 'fragmentation' to exclusively describe significant breakdown of a grain into large pieces (e.g., (Miller and Jerolmack, 2021)). All of these processes act on sediment particles while they are being carried by water current in rivers, leading to a reduction of their size and alteration of their shape. Researchers have evidenced a general relationship between roundness and attrition of sediment transported as bed-load in rivers (Novák-Szabó et al., 2018). The effectiveness of attrition/abrasion has been shown to vary with the lithology of the clasts (Attal and Lavé, 2009). The varied grade of weathering of a source rock prior to its introduction as sediment into the fluvial environment can lead to varied degree of roundness for the same transport distance (Gale, 2021).

There are different views regarding the controls on and trends in pebble roundness as one moves downstream. Figure 1 illustrates the downstream change in the perimeter shape of the pebbles. Wentworth (1922) studied the relationship between pebble shape and flow distance based on roundness and flatness ratios using field and laboratory measurements, and suggested the rounding effect of abrasion diminishes downstream. Field-measured roundness shows a systematic trend, i.e., an increase in rounding with distance downstream, in the upstream part of catchments where there is no contribution of more angular pebbles from lateral tributaries (Brewer and Lewin, 1993; Roussillon et al., 2009). Similarly, a two-phase evolution of pebble shape was proposed by Miller et al. (2014) based on theory and measurements of pebble roundness along a river in Puerto Rico. In the first phase (headwaters), pebbles are rapidly rounded, while in the second phase (downstream part of the river system), pebbles are reduced in size with little changes in roundness. Roussillon et al. (2009) argue that the roundness trends can persist over long distances (at least 20-50 km) and disagree with the idea that roundness changes are limited to the upmost part of the fluvial network. All these studies (Roussillon et al., 2009; Wentworth, 1922; Vanbrabant et al., 1998; Miller et al., 2014) suggest a non-linear relationship between transport distance and roundness; grains round rapidly in the first part of their journey through the fluvial system, and this rounding slows further downstream.

Although the three parameters - shape, size and roundness - are usually associated with each other, there is debate about how they co-evolve and whether some of these dominate (Domokos et al., 2014). The downstream evolution of a pebble's shape and roundness has been shown to be controlled by the initial grain size, hardness and existence of fabrics within the rock (Kuenen, 1956; Lindsey et al., 2005), with some of these factors directly related to the lithology of the pebble itself (Kuenen, 1956; Sneed and Folk, 1958; Kodama, 1994; Sklar and Dietrich, 2001). For example, pebbles of limestone and andesite achieve their maximum roundness after only a few kilometres of transport, whereas rocks with high silica content like chert and quartzite can still have low relative roundness after tens of kilometres of transport (Sneed and Folk, 1958). Due to the brittle nature of rocks with high silica content, roundness may not change or may even decrease during transport because of spallation or fracturing (Sneed and Folk, 1958). This may also happen in polycrystalline rocks like pegmatite (igneous rocks with mineral grains > 5 cm) due to physical breakage along large grain boundaries during transport (Lindsey et al., 2005).

A study in an Alpine river also showed that the water discharge and flow strength do not exert the main control on the shape and size of fluvial pebbles (Litty and Schlunegger, 2017). Instead, the lithological composition of the pebbles itself, and therefore that of the sediment supplied to the river through mass failure, was the key determining factor on the pebble shape and roundness (Litty and Schlunegger, 2017). This result is consistent with the study by Attal and Lavé (2009) who designed a circular flume to replicate the abrasion processes effective during vigorous fluvial transport in powerful Himalayan rivers during the monsoon (Attal et al., 2006): while the experiments at high flow velocities caused widespread pebble breakage leading to abrasion rates an order of magnitude greater than previously published, breakage did affect mostly schist and sandstone pebbles (Attal and Lavé, 2009). Abrasion rates for other lithologies such as granite, gneiss and limestone remained comparable to previously published results, consistent with field observations (Attal and Lavé, 2006).

## 1.2 A brief history of shape indices and measurement techniques

Although shape is a fundamental property for all kind of objects, including sediments, it remains one of the most difficult to characterize and quantify (Wentworth, 1933; Barrett, 1980; Blott and Pye, 2008). Different terms such as 'form' (Sneed and Folk, 1958), 'roundness' (Wentworth, 1922, 1923), 'sphericity' (Wadell, 1935) and 'irregularity' (Blott and Pye, 2008) are most commonly used to define the shape of sediment particles. The term sphericity is often used synonymously with roundness. Wadell (1935) first proposed the term sphericity, which represents the degree to which a particle approximates the shape of a sphere, and is independent of its size. In contrast, roundness refers to the sharpness of pebble edges (Cruz-Matías et al., 2019). Even though the concepts of roundness and sphericity are related, they are two distinct terms. For example, an object with a regular dodecahedron shape has a high degree of sphericity but has very low roundness (Blott and Pye, 2008). There are numerous methods available for the calculation of roundness, with new methods still being proposed and old methods falling out of favour. Some earlier definition of roundness include: the ratio of the radius of curvature of the sharpest corner to the mean radius of the particle (Wentworth, 1922), the ratio of the diameter of curvature of the sharpest corner to the intermediate axis of grain (Kuenen, 1956), the ratio of the diameter of curvature of the sharpest corner to the diameter of the largest inscribed circle (Dobkins and Folk, 1970), as well as indices based on the ratio of a pebble's perimeter to its area in 2D (Roussillon et al., 2009), to name only a subset of previously applied metrics.

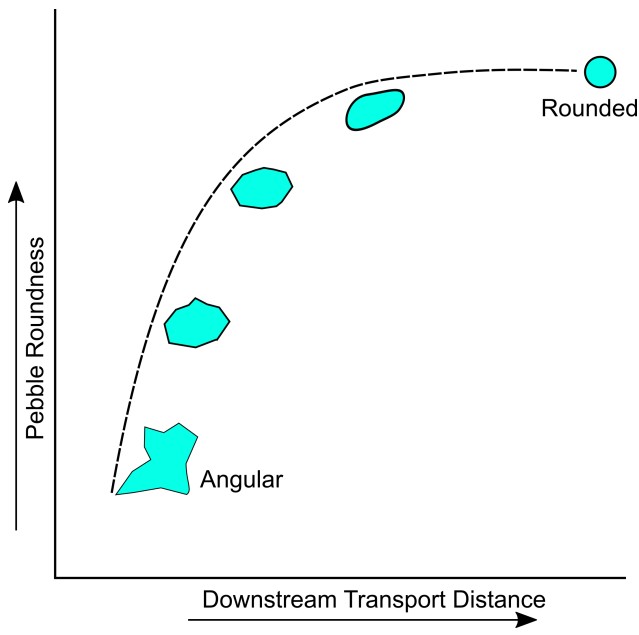

**Figure 1.** Schematic diagram showing the downstream shape and size changes of a particle as it is transported along a river. An angular pebble becomes rounded after travelling a certain distance along the river. Note that this diagram shows a conceptual model in which the relationship between roundness and transport distance is non-linear, as suggested by previous studies (see text in Sect. 1.1).

Many studies are either based on direct measurements in the field (e.g., (Wentworth, 1922, 1923; Litty and Schlunegger, 2017)) or the manual tracing of outline of pebbles using the 2D images (e.g., (Quick et al., 2019)). Both of these methods are subject to human bias and are almost impossible to replicate. However, studies developing an automated workflow to reduce the subjectivity in calculating the shape parameters have been recently published (e.g., (Roussillon et al., 2009; Cassel et al., 2018; Bodek and Jerolmack, 2021)). Roussillon et al. (2009) developed a tool for the automatic extraction of pebble shape

from 2D images. Similarly, Cassel et al. (2018) assessed and validated the use of an automated toolbox to define the relation between roundness metric trends and abrasion using the 2D images. Tunwal et al. (2020) proposed image-based automated particle shape analyses for both consolidated and loose sediments which can measure traditional, mathematically complex and common geometric shape parameters. Thus, advances in technology are making automatic extraction of shape parameters possible. Automated image analysis and Fourier grain shape analysis allow modern workers to analyse a high volume of

roundness metrics quickly and easily (Diepenbroek et al., 1992).

    However, while a good method should have a high degree of reproducibility, the choice of an appropriate roundness metric is still based on the judgement of authors, leading to difficulties when comparing results from different studies (Barrett, 1980; Diepenbroek et al., 1992). To address this problem, some researchers (Roussillon et al., 2009; Purinton and Bookhagen, 2019; Detert, 2020; Steer et al., 2022) have developed automated methods to extract multiple indices characterizing a pebble's shape

and size, which facilitates comparison and correlation of roundness indices obtained using different methods. For instance,

Roussillon et al. (2009) compared a series of geometric parameters that characterize roundness, such as Wadell (1935)'s roundness index ($r_w$), Durian et al. (2006)'s roundness index ($r_d$) and Cottet (2006)'s roundness index ($r_p$) along with the pebbles' axial ratio ($a/b$), circularity and convexity. What these studies showed is that even with automated methods, the above indices are sensitive to the method of assessments.

## 1.3 Motivation

This study has both a methodological and research-focused aim. We provide a methodology for the measurement of pebble roundness, which is automatic, time efficient, and provides results that can be replicated when applied to the same image. While Roussillon et al. (2009) and Cassel et al. (2018) describe automatic methods of pebble shape analysis using 2D images, we provide additional information on site location, pebble collection and photography, as well as details about an image processing technique using an open access graphics user interface (GUI) software. We present a workflow to help researchers replicate results and adopt this technique in future studies. We use the Isoperimetric Ratio as the geometric parameter to characterize a pebble's roundness, building on previous recent studies (Miller et al., 2014; Szabo et al., 2015; Quick et al., 2019); more specifically, we use the Normalized Isoperimetric ratio ($IR_n$) first developed by Quick et al. (2019) (see next section for full justification of this choice). Based on measurements in two Himalayan catchments with varied rock types and provenance settings, we propose a new model to relate the roundness ($IR_n$) with the transport distance ($d$), that is, the distance travelled by the pebbles from their entrance point in the river system to the location where they were measured. Details of our roundness model, which mirrors Sternberg's law of mass loss (Sternberg, 1875), are provided after presentation of the roundness data collected along the two Himalayan rivers, as these data are needed to contextualise the model. We further explore the applicability of our roundness-distance relationships to estimate the distance travelled by Miocene and Pliocene sediments in the Himalaya.

## 2 Materials and Methods

In this section, we describe the shape index chosen in this study, as well as a complete workflow for the field data collection, image processing and data analysis, including the description of the study catchments and collection strategy.

### 2.1 Choice of shape index

As mentioned in the introduction section (Sect. 1.2), many shape indicators exist. Cassel et al. (2018) explored the effect of pebble position, image resolution and enhancement on the indices of roundness and shape measurements in which the outline of pebbles is extracted from 2D images using the automated tool developed by Roussillon et al. (2009). Indices such as $r_p$, convexity and circularity were found to be more powerful than $r_w$ to assess roundness (Roussillon et al., 2009). However, the indices $r_p$ and $r_d$ were found to have different sensitivities to the image resolution and enhancement. Hence, circularity appears to be the better choice to measure pebble roundness when the outline of a pebble is extracted using 2D image processing. A series of recent studies have used the 'Isoperimetric Ratio', a parameter equivalent to circularity, to measure the downstream

evolution of pebble roundness in the fluvial environment (Miller et al., 2014; Szabo et al., 2015; Quick et al., 2019; Bodek and Jerolmack, 2021).

The 'Isoperimetric Ratio' is equivalent to the term 'roundness' first proposed by Cox (1927) (using a different name) which is defined as the ratio of the area ($A$) of a pebble in 2D to its perimeter ($P$), as shown in Equation 1 (Blott and Pye, 2008). The theoretical value of the 'Isoperimetric Ratio ($IR$)' varies between 0 and 1; a perfect circle (perfectly spherical pebble) will have a value of 1, with measured $IR$ values for natural pebbles typically ranging from 0.5 to 1.0 (Quick et al., 2019).

$$IR = (4\pi A)/P^2 \tag{1}$$

However, Roussillon et al. (2009) and Quick et al. (2019) found that $IR$ is sensitive to both elongation and roundness. For instance, a perfectly rounded elliptic pebble with an axis ratio $b/a$ of 0.5 (where $a$ and $b$ are the longest and shortest axes in the plan considered, respectively) will have an $IR$ of 0.84, which could be similar to that of a more angular but spherical pebble ($b/a = 1$) (Figure 2). Quick et al. (2019) found that the maximum $IR$ a pebble can achieve decreases with decreasing axis ratio. They developed a 'Normalized Isoperimetric Ratio' ($IR_n$) designed to remove any dependency on elongation, and only measure the angularity (or roundness) component from the $IR$ (Equation 2). The normalized isoperimetric ratio ($IR_n$) is calculated by dividing $IR$ by the maximum theoretical isoperimetric ratio ($IR_t$) a pebble can achieve based on its measured $b/a$ ratio. The Ramanujan approximation (Villarino, 2005) is used to calculate the area and perimeter of an ellipse of axes $a$ and $b$ for the calculation of the maximum theoretical isoperimetric ratio (Equation 3).

$$IR_n = IR/IR_t \tag{2}$$

$$IR_t = \pi(a+b)(1 + (3h/(10 + (4 - 3h)^{1/2}))) \tag{3}$$

where

$$h = (a - b)^2/(a + b)^2 \tag{4}$$

Our subsequent analyses use $IR_n$ which removes the effect of elongation and gives the true measure of roundness of the pebbles.

## 2.2 Study catchment and site selection

Our aim in this study is to quantify the degree to which pebbles round as they travel downstream. It is therefore essential to select catchments where an identifiable lithology is only supplied from one portion of the catchment so we can be confident rounding measurements taken as a function of transport distance are not confounded by addition of less rounded clasts of the same lithology further downstream. We have identified two catchments that meet this criterion, from which our samples are

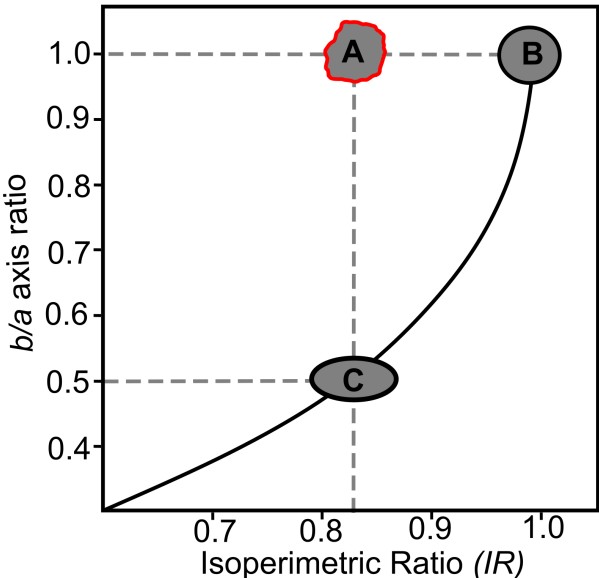

**Figure 2.** Illustration of the effect of elongation ($b/a$ axis ratio) on the roundness measurement (Isoperimetric Ratio). In the figure, A is an angular ($IR = 0.84$) and spherical ($b/a = 1.0$) pebble, B is a perfectly rounded ($IR = 1.0$) and spherical pebble ($b/a = 1.0$), and C is a perfectly rounded ( but $IR \neq 1.0$) and elliptical ($b/a = 0.5$) pebble. Although the elliptical pebble (C) is perfectly rounded, its roundness is equivalent to that of the angular and spherical pebble (A) due to elongation. The solid black line in the figure represents the theoretical maximum isoperimetric ratio as a function of the axis ratio. With the use of the Normalized Isoperimetric Ratio (the roundness metric used in this study), pebbles B and C will have the same roundness, thus removing the effect of elongation and measuring only the roundness (Figure adapted from Quick et al. (2019)).

drawn. The catchments lie in western and central Nepal and their rivers are the Banganga River and Rapti River, respectively
(Figure 3). Both of these catchments have headwaters in Lesser Himalayan lithologies; neither are connected to large trans-
Himalayan rivers upstream of our sampling areas.

The Banganga River contains two thick ($\sim$100 m) quartzite units near the headwaters of the catchment. The remainder of the
catchment consists of Precambrian-Paleozoic meta-sedimentary rocks and Eocene-Pleistocene sedimentary rocks (Sakai, 1983;
Dhital, 2015) (Figure 3 (c)). Unlike many locations along the Himalayan mountain front, there are no molasse conglomerates
that may input recycled pebbles into the river (Quick et al., 2019). Thus the Banganga River is perfectly suited for studying the
rounding of quartzite pebbles over a known distance from their source.

The Rapti River catchment comprises an exposed granite body in its headwaters, and so is suited to measuring the progressive
rounding of granite pebbles downstream from their source (Figure 3 (d)). Although there is both quartzite and granite in the
upstream reaches, quartzite bands are also exposed downstream. Moreover, conglomerate beds of Upper Siwaliks are also
exposed in the southernmost part of this catchment, and so recycling may be an issue (Quick et al., 2019). Hence, we only

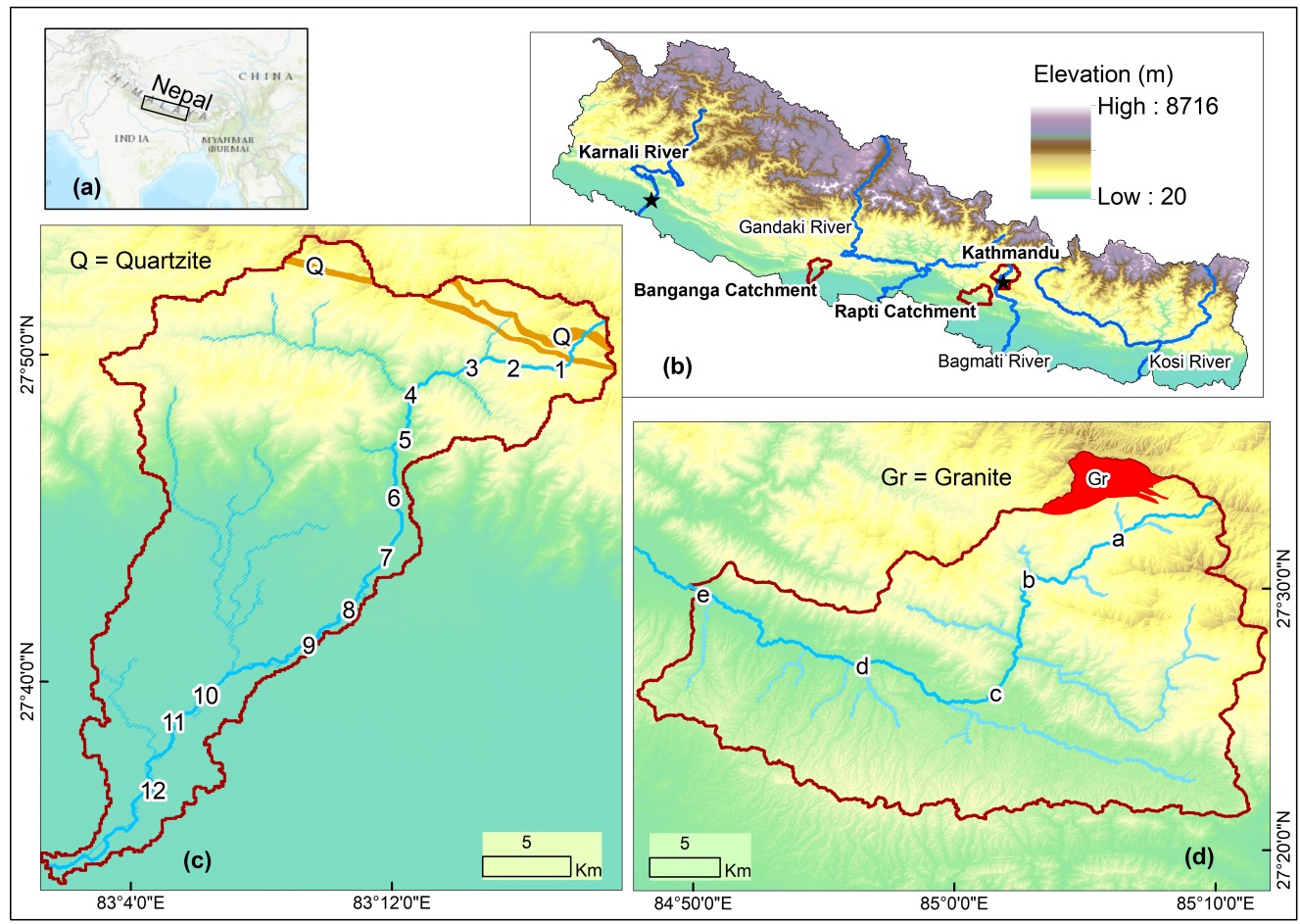

**Figure 3.** Location map. **a**: Location of the Himalaya and Nepal. **b**: Political boundary of Nepal with the location of river catchments used for sample collection. **c** and **d**: study catchments and river networks showing the location of each sample site and bedrock exposure area for the lithology of the pebble collected in the field along the Banganga River (**c**) and Rapti River (**d**). Note: 1, 2, ..., 12 in (**c**) are the sample sites for quartzite pebbles and a, b, ..., e in (**d**) are the samples sites for granite pebbles. In both catchments, transport distance to each sample site is calculated from the channel head using Mudd et al. (2022). The black stars in (**b**) represent the sampling sites for pebbles and clasts from the Karnali River and the Kathmandu Basin, as discussed in Sect. 4. Data source: Global map- ESRI Basemap, Topographic data- open topography ALOS World 3D (Japan Aerospace Exploration Agency, 2021), Political boundary of Nepal and rivers- Department of Survey-Nepal, Lithological boundaries- Sakai (1983) and Dhital (2015).

.

consider granite pebbles for the roundness analysis to avoid multiple sources and recycled pebbles of quartzite in the modern sediment of the Rapti River.

At first, sites for the pebble collection were identified using open access global base maps (Google Map and Google Earth). Where possible, we aimed to have relatively uniform spacing in terms of downstream flow distance between sampling sites. Some variation in sampling distance along the river does occur due to site accessibility.

The quartzite pebbles were collected at twelve locations along the Banganga River and granite pebbles were collected from five locations along the Rapti River. The location of each sampling site is shown in Figure 3. We extracted flow distance from the channel head using the LSDTopoTools software (Mudd et al., 2022). We then compared the normalized isoperimetric ratio against flow distance for our samples. The total distance covered by the pebble collection along the river is ∼50 km in each catchment. We only sampled active gravel bars from the main channel, and we avoided gravel bars with any evidence of human influence. The most common influences observed in the field were mounds of clasts indicative of mining for aggregate, and construction of temporary diversions or access roads along and across the river channel. Additionally, we did not sample gravel bars close to landslides.

In addition to these two catchments, we use roundness data from two other locations (marked by stars in Figure 3 (b)) along the Karnali River in western Nepal and the Kathmandu Basin in central Nepal. Based on these data, we discuss the applicability of our new roundness model for both modern and ancient river systems in Sect. 4. The data for the stratigraphic record of the Kathmandu Basin was collected during this study, and data on sediment recycling along the Karnali River was previously collected by Quick (2021).

## 2.3 Pebble Collection and photography in the field

The Banganga River covers the part of synclinorium in western Nepal mapped by Sakai (1983). This catchment includes quartzite bands exposed at two locations towards the channel head. Downstream from the lower quartzite band, only shale, limestone, dolomite, sandstone, mudstone and siltstone are exposed. The quartzite bands are competent and white to grey in colour, and are the source of boulders, cobbles and pebbles that can be distinguished in gravel bars several kilometres downstream. This area lacks granite in the source region, so only the pebbles of quartzite rock are collected from this catchment. We carefully examined pebbles based on texture and mineralogy using a hand lens, thus we are confident all our sampled pebbles in the Banganga River are indeed quartzite and not some other rock type.

We applied a similar sampling procedure in the Rapti catchment, south of the Kathmandu. The field identification of granite pebbles is easier than identification of quartzite pebbles as there are no other rocks with igneous texture exposed in this study catchment. The location of the quartzite band in the Banganga catchment, and the granitic body in the Rapti catchment, are shown in Figure 3.

Upon arriving at a potential sampling site, we first assessed whether the gravel bar was close to the active channel (i.e., not a terrace). We then extended a 25 m linear transect along the gravel bar and sampled all pebbles of quartzite and/or granite that could be lifted manually by both hands. In upstream regions, close to the channel head, we found it difficult to find enough grains of a size that could be lifted from the gravel bar. Therefore, multiple (a maximum of three) 25 m transects were used for the pebble collection in sampling locations with very coarse (up to boulder size) sediments. To minimise the bias in sample collection, only the pebbles from a single gravel bar were collected at each sampling site. We aimed to collect approximately

100 pebbles at each site, but there were sites where we collected fewer than 100. In total, we collected and photographed about 2000 pebbles in the field.

Our method requires imaging the pebbles to extract their outlines, so we needed a method of collecting high contrast images in the field. Studies have attempted to collect similar images. For example, Cassel et al. (2018) used a flat surface of 1 m$^2$ with a red background to photograph the pebbles in the field. Here, we covered a rigid board with a black blanket and used this surface to photograph the pebbles. We placed pebbles along their longest and intermediate axes (covering the maximum surface area). We then held a camera directly above the surface at chest height, and took a photograph of the pebbles in a scene that included a scale of known dimension. We used a large umbrella to prevent the photographed surface from being exposed to direct sunlight. This eliminates the shadow of pebbles and mitigates any reflecting surfaces of multi-coloured mineral grains within the granitic pebbles. We took care to remove dust particles and any other field dirt from the flat surface.

## 2.4 Image Processing

Many previous studies of pebble roundness are based on manual digitisation of pebble outlines from photographs of gravel bars (Quick et al., 2019; Miller et al., 2014). The results from manual tracing are difficult to replicate and uncertainties are introduced because of personal judgement. Additionally, manual tracing is a time consuming process. We use image processing to circumvent these issues (Figure 4). Our workflow extracts the area ($A$), perimeter ($P$), the length of the major axis ($a$) and minor axis ($b$) using the automatic digitisation of a pebble's outline. There are challenges to automate the pebble extraction process including the overexposure, shadowing, wet pebble and bleeding effects in the 2D images of pebble (Cassel et al., 2018). Hence, the automated outline extraction demands a multi-stage technique to measure the geometric parameters of pebble roundness from the 2D images (Figure 4).

The basic principle followed in this study is to read the pebble silhouette automatically by a software using a colour threshold. The colour threshold value differentiates the black background and the pebble area. After this step, we removed the background surface and the pebble's outline was extracted to measure the geometric parameters. Roundness values vary with the orientation of the object in a raster environment; indeed, an image in a raster will have pixelated contours, and a line that is oriented NW-SE will be 1.4 times longer than the same line oriented N-S, due to the tracing of the line following the pixel contours and adding distance when the line is not perfectly oriented in the direction of the grid (N-S or E-W); this occurs irrespective of the resolution. We address this issue by converting the pebble outline from 2D image raster to the vector format. We also perform a smoothing of the pixel boundaries while converting the pixel outline into a vector outline. The smoothing is done in such a way that the polygons contain a minimum number of segments while remaining as close as possible to the original raster cell edges. The methodology that we adopt for the 2D image processing using the ImageJ (Schneider et al., 2012) and ArcGIS is described in detail below.

To convert 2D field photographs (from Sect. 2.3) into pixels, open an image in ImageJ and convert it to an 8-bit type image. Use the "set scale" option by drawing a line along the object of known dimension and providing the dimension with a unit. Then adjust the image based on the threshold value using "Adjust threshold". This is an important step that extracts the shape of the pebbles by separating the pixels into foreground (object of interest - pebbles) and background (everything else). After

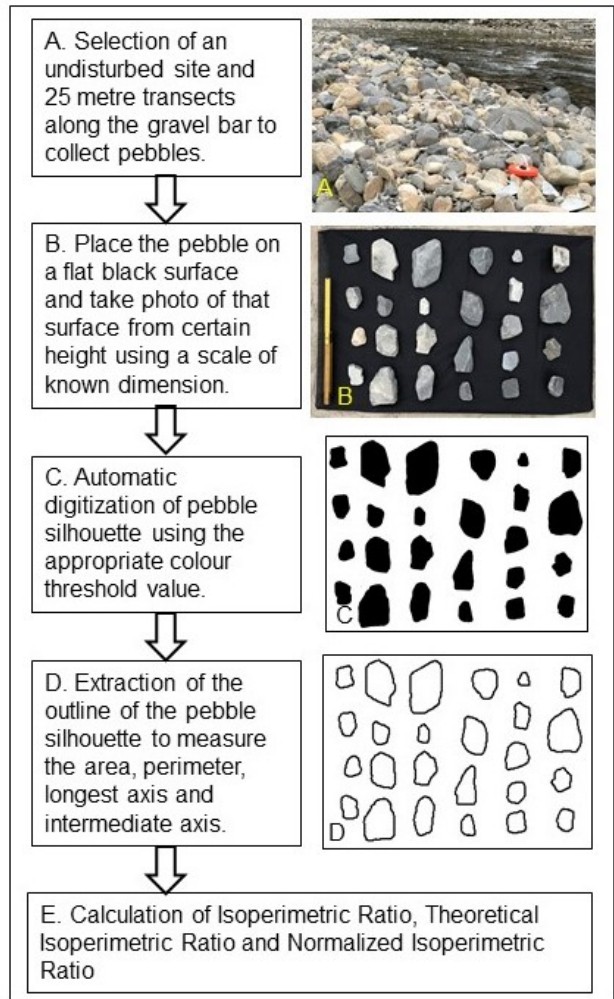

**Figure 4.** Flow chart showing the work flow for the site selection, pebble collection, photography technique, image processing and calculations.

adjusting the threshold value, the image updates in real-time to show the pixels included in the foreground and background. The image is then converted into a binary image and the 'Fill hole' option is applied to fill any holes or gaps in the foreground objects.

Once a satisfactory binary image is created in ImageJ, export the image as a GeoTiff file that can be imported in GIS
environments. To measure the area, shape, longest axis, and intermediate axis, open the GeoTiff image in ArcGIS and provide a reference in the meter system. Then use "Raster calculator" to convert the GeoTiff image into a raster integer file, which allows opening of the attribute table of the raster file. Open the attribute table, add a field (e.g., "pebble-shape") and assign 1 for the pixels comprising pebbles and 0 for the background pixels. Then save and stop editing.

Next, use the "Raster to polygon" conversion tool to convert the raster file into vector format, selecting the "pebble-shape" column in the value field that was added earlier. Also, select the "Simplify polygon" option to eliminate the pixelisation effect in the area and perimeter measurements. This step creates the polygon for the pebble shape, and the area and perimeter of the shape are calculated using the "calculate geometry" function in the attribute table. Finally, the major ($a$) axis and intermediate ($b$) axis are measured using the "Minimum bounding geometry" function from the search box tool in ArcGIS, with "Geometry type" as convex hull and "Geometry characteristics" as attribute added. This step provides all the measurements necessary to calculate the parameter used as a measure of roundness in this study. Although we used ArcGIS for the conversion of raster images into vector polygons, the work could equally be done using other open-access GIS software/packages.

## 3 Results

### 3.1 Downstream Changes in Roundness and New Roundness Model

We calculate $IR_n$ for each pebble from all locations for the quartzite and granite pebbles (see box plot in Figure 5 (a) for granite pebbles and Figure 5 (b) for quartzite pebbles). The range of $IR_n$ values is wider at the upstream sites and narrows down at the downstream sites, particularly for the granite pebbles. Each location consists of a mix of angular to rounded pebbles. For example, at the upstream sites there are a small number of pebbles that are as round as pebbles that have travelled $\sim$ 50 km downstream. Because each site has a mixture of roundness values, we have calculated five different percentiles to capture the range of and changes in $IR_n$ at each site ($5^{th}, 25^{th}, 50^{th}, 75^{th}$ and $95^{th}$ percentiles). These percentiles represent the angular to rounded sub-populations of pebbles from each location. The $IR_n$ value of the $5^{th}$ percentile represents the most angular pebbles and the $95^{th}$ percentile represents the most rounded pebbles in that particular location.

As an exploratory analysis step, we applied a linear regression to each set of percentile data as a function of downstream flow distance (Figure 5 (c) and (d)); while we expect $IR_n$ to display an asymptotic behaviour towards $IR_n = 1$, we find that the trends can be reasonably approximated by linear fits over distances of $\sim$50 km. All trends show that the roundness of every percentile, including the median, increases downstream (Figure 5 (c) and (d)).

We make two key observations, that support the development of our new rounding model:

– Granite pebbles are rounder than quartzite ones when comparing the percentiles across lithologies.

– The linear fits of all percentiles have a comparable slope for the quartzite pebbles, but the slope decreases rapidly with increasing percentile (that is, slope is lower for the most rounded populations as $IR_n$ approaches 1) for the granite pebbles.

If we make the assumption that if two grains have the same roundness, they will round at the same rate, then we can infer that each percentile represents a population evolving downstream. The linear fits in the graph could therefore represent sections of an asymptotic trend occurring over much longer distances, with a gradient that decreases as $IR_n$ approaches the asymptote (see Figure 6).

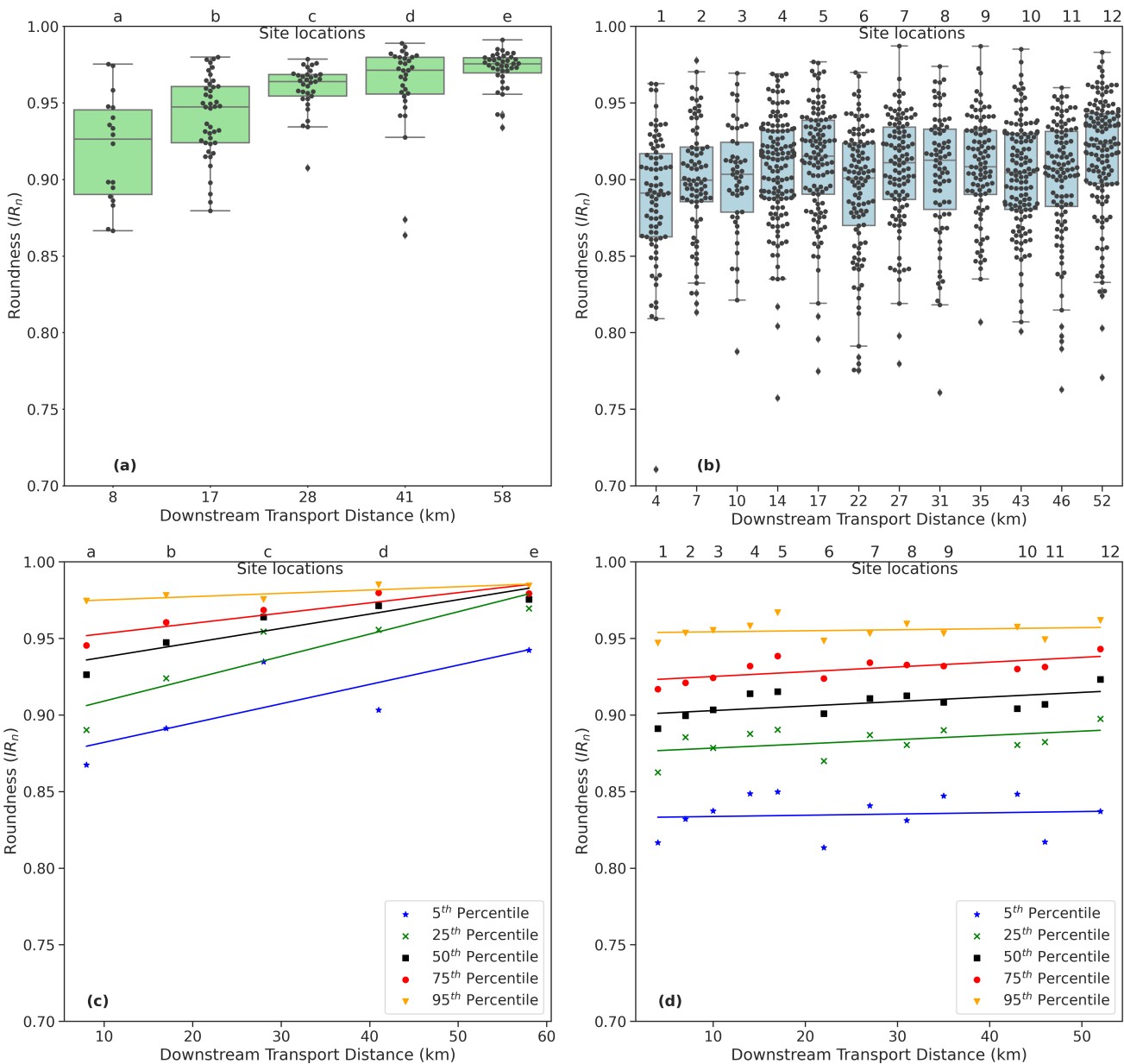

**Figure 5.** Box plots of the raw data showing the range of roundness values at each location for (**a**) granite pebbles and (**b**) quartzite pebbles. The bottom row shows linear fit to the downstream percentiles ($5^{th}$, $25^{th}$, $50^{th}$, $75^{th}$ and $95^{th}$) of (**c**) granite pebbles from Rapti River and (**d**) quartzite pebbles from Banganga River. Note that the axes of the top figure is categorical such that the box plots are uniformly spaced and are not positioned by downstream distance. a, b, ..., e in (**a**) and (**c**) are the samples sites for granite pebbles, while 1, 2, ..., 12 in (**b**) and (**d**) represent the sample sites for quartzite pebbles (for locations, see Figure 3 (c) and (d)).

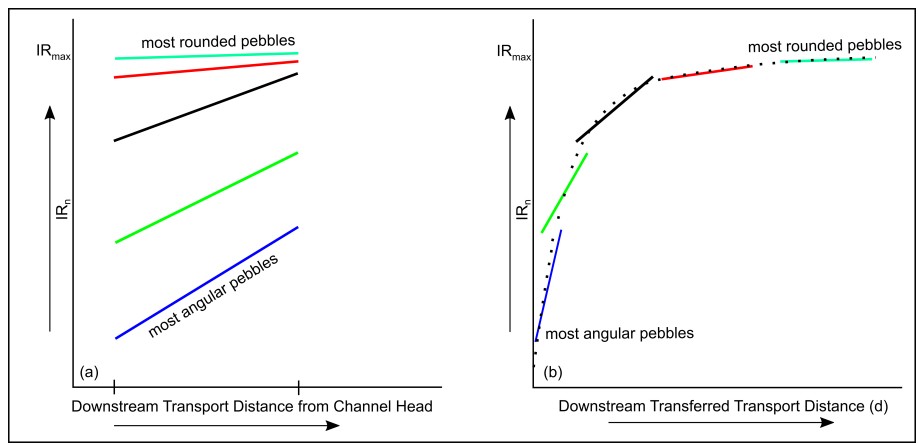

**Figure 6.** Schematic diagram showing our conceptual roundness model, including the idea that linear fits to each percentile data over a distance of 50 km represent the evolution of roundness for pebble populations starting their journey with different roundness values, and therefore represent various segments of the complete asymptotic roundness curve, with slope decreasing rapidly as $IR_n$ approaches 1 (**a**). Reconstructing the complete roundness curve can be achieved by shifting each percentile data by an increasingly greater distance with increasing percentile (and therefore roundness) (**b**). See text (Sect. 3.2) for description of the approach developed to determine the best fit shifting distances.

As rounding is driven by abrasion, we propose a rounding model that mirrors Sternberg's law of abrasion which predicts that a pebble's size or mass will decay exponentially downstream under the effect of abrasion (Sternberg, 1875). This model is consistent with previous studies that have shown that there is a nonlinear relationship between transport distance and pebble roundness (Wentworth, 1923; Miller et al., 2014). We begin by assuming that this nonlinear relationship extends to the relationship between $IR_n$ and transport distance, and that the maximum $IR_{max}$ value that will be asymptotically approached

is unity. In other words, a pebble will eventually, given enough transport distance, become perfectly rounded. Based on these assumptions, we propose the following equation for the evolution of $IR_n$ downstream:

$$IR_n = IR_{max} - ke^{-\lambda d} \tag{5}$$

where $IR_{max}$ is the maximum roundness value that the pebble can achieve, which is theoretically 1, $d$ is the transport distance, $k$ is a prefactor value that controls the initial roundness of the pebble, and $\lambda$ is a coefficient that defines the rate at

which the pebbles round as a function of transport distance.

The advantage of this equation is that the coefficients $\lambda$ and $k$ can be obtained through linear regression of field data. The equation can be rearranged as follows:

$$\ln(IR_{max} - IR_n) = \ln(ke^{-\lambda d}) \tag{6}$$

As $IR_{max} = 1$, the equation becomes:

$$\ln(1 - IR_n) = \ln(k) + \ln(e^{-\lambda d}) \tag{7}$$

$$\ln(1 - IR_n) = \ln(k) - \lambda d \tag{8}$$

In a plot of $ln(1 - IR_n) = f(d)$, the slope of the linear regression is $-\lambda$ and the intercept $ln(k)$. In Sect. 3.2, we describe how we process our data to derive the theoretical rounding curves and coefficients for our granite and quartzite pebbles.

## 3.2 Derivation of Rounding Curves and Coefficients for our Granite and Quartzite Pebbles

As mentioned earlier, we propose that the distance ($\sim$50 km) we covered in the field to collect the pebble roundness data is not sufficient to generate the complete roundness curve. We also propose that each percentile represents a given population of pebbles starting its journey with a given roundness, and that each fit to the percentile data over 50 km represents a section of the complete roundness curve (Figure 6). Segments corresponding to greater percentiles have higher roundness values and lower slopes (in particular for the granite), and would therefore correspond to parts of the curve closer to the asymptote, i.e.,

further to the right (greater transport distance) in Figure 6. The challenge is to determine the transport distance by which each percentile data has to be shifted to the right to reconstruct the complete roundness curve.

The model we propose predicts a linear relationship between transport distance ($d$) and $ln(1 - IR_n)$. We calculate $ln(1 - IR_n)$ for all our field data, and run a sequential analysis to estimate by how much each percentile data have to be shifted in terms of distance to produce the best linear fit of $ln(1 - IR_n) = f(d)$ for each lithology. We begin by plotting $ln(1 - IR_n) = f(d)$ for the

320 $5^{th}$ percentile data for all field locations. We then add the $ln(1 - IR_n)$ values for the $25^{th}$, $50^{th}$, $75^{th}$ and $95^{th}$ percentile data with the transport distance shifted by varying amounts further downstream; we call this shifted transport distance 'transferred distance'.

**Table 1.** Transferred distances for each percentile that gives the best fit using the down hill gradient for granite pebbles.

| Rock type: | | Granite |
|---|---|---|
| Location: | | [a, b, c, d, e] |
| Field distance (km): | | [8, 17, 28, 41, 58] |
| Percentile | Transferred distance (km) | New distances (km) |
| $5^{th}$ percentile | 0 | [8, 17, 28, 41, 58] |
| $25^{th}$ percentile | 21 | [29, 38, 49, 62, 79] |
| $50^{th}$ percentile | 37 | [45, 54, 65, 78, 95] |
| $75^{th}$ percentile | 51 | [59, 68, 79, 92, 109] |
| $95^{th}$ percentile | 80 | [88, 97, 108, 121, 138] |

We use an optimisation technique to find the best-fitting linear regression model for a set of data points. The data consists of distance data $X5$, $X25$, $X50$, $X75$, and $X95$, and the associated $ln(1 - IR_n)$ data corresponding to each percentile ($5^{th}$,

**Table 2.** Transferred distances for each percentile that gives the best fit using the down hill gradient for quartzite pebbles.

| Rock type: | | Quartzite |
|---|---|---|
| Location: | | [1, 2, 3, 4, 5, 6, 7, 8, 9, 10, 11, 12] |
| Field distance (km): | | [4, 7, 10, 14, 17, 22, 27, 31, 35, 43, 46, 52] |
| Percentile | Transferred distance (km) | New distances (km) |
| $5^{th}$ percentile | 0 | [4, 7, 10, 14, 17, 22, 27, 31, 35, 43, 46, 52] |
| $25^{th}$ percentile | 144 | [148, 151, 154, 158, 161, 166, 171, 175, 179, 187, 190, 196] |
| $50^{th}$ percentile | 245 | [249, 252, 255, 259, 262, 267, 272, 276, 280, 288, 291, 297] |
| $75^{th}$ percentile | 363 | [367, 370, 373, 377, 380, 385, 390, 394, 398, 406, 409, 415] |
| $95^{th}$ percentile | 553 | [557, 560, 563, 567, 570, 575, 580, 584, 588, 596, 599, 605] |

$25^{th}$, $50^{th}$, $75^{th}$ and $95^{th}$). The distance $X5$ corresponding to the $5^{th}$ percentile is kept as the original values from the field. The primary objective of the approach is to determine the optimal distances ($X25$, $X50$, $X75$, $X95$) that yield the best linear fit by transferring the roundness data from the $25^{th}$, $50^{th}$, $75^{th}$ and $95^{th}$ percentiles further downstream, thereby increasing transport distance $d$. We consider the $R-squared$ values (with vertical residue) as the evaluation metric. The term 'transferred transport distance' is a downstream distance along which the percentiles higher than the $5^{th}$ percentile are shifted to a greater distance, assuming that the higher percentile represents pebbles transported to that greater distance. Hence, this distance does not represent the distance from the channel head but instead represents the required transported distance for pebbles to achieve greater roundness, beginning from an initial roundness at the distance $d = 0$. In this model, this initial roundness is set by the lowest percentile data ($5^{th}$ percentile) for which distance has not been shifted and the prefactor $k$.

The optimisation process used is the downhill gradient method, implemented through the 'scipy.optimise.minimise' function using 'Nelder-Mead' method (Nelder and Mead, 1965). This method aims to minimise the negative of the performance metric, effectively maximising the $R-squared$. The process starts with initial parameter values and iteratively adjusts these parameters to optimise the $R-squared$. The 'Nelder-Mead' method is selected as the optimisation algorithm due to its effectiveness in handling non-linear optimisation problems without requiring gradient information.

The function at first calculates the $R-squared$ value for the set of parameters ($ln(1-IR_n)$ and $d$) and appends the $R-squared$, slope, and intercept values. It then proceeds to optimise the parameters using the 'minimise' function. Once the optimal parameters are determined, new distance values ($X25$, $X50$, $X75$, $X95$) are calculated using the optimised offsets. The new distances after the transformation of percentile is shown in Table 1 for granite pebbles and Table 2 for quartzite pebbles. The linear regression model is then fitted to the concatenated data ($X = X5$, $X25$, $X50$, $X75$, $X95$ and $Y = ln(1-IR_n)$ from the $5^{th}$, $25^{th}$, $50^{th}$, $75^{th}$, and $95^{th}$ percentiles, respectively) using the linear regression model. The value of the pre-factor $k$ derived from this optimisation technique is 0.145 and 0.174 for the granite and quartzite pebbles, respectively (Figure 7). The rounding coefficient $\lambda$ is 0.018 and 0.002 for the granite and quartzite pebbles, respectively; the granite's $\lambda$ is nine times that of quartzite. Based on these parameters, we can construct the theoretical roundness curve over longer distance for these two

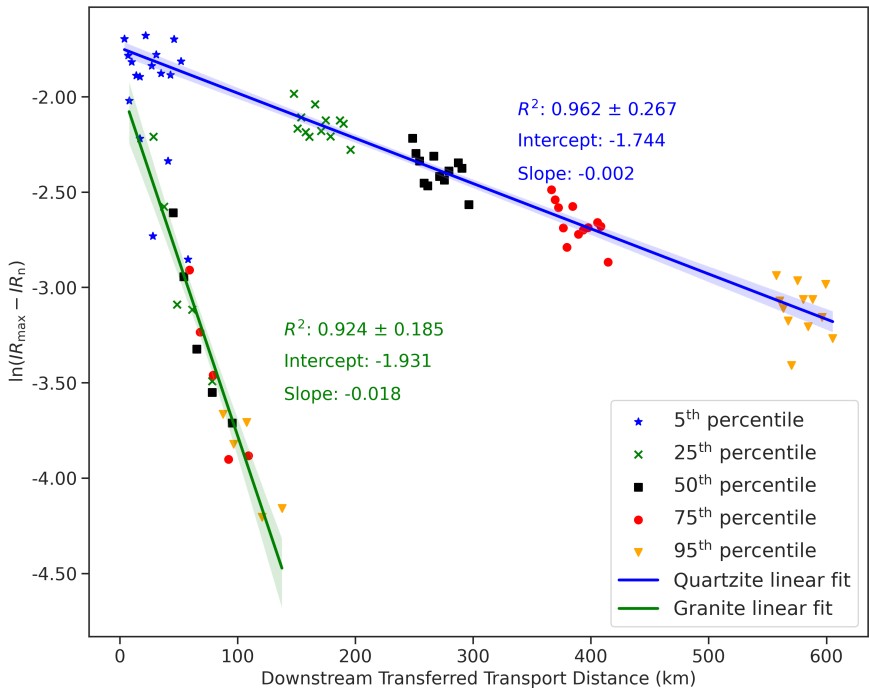

**Figure 7.** Plot of $\ln(IR_{max} - IR_n)$ against downstream transferred transport distance for granite pebbles and quartzite pebbles. Each marker and colour represents percentile roundness data. The blue and green lines represent the best linear fits of transferred percentile roundness data for quartzite and granite, respectively. The shaded area around the regression line in the plot represents the 95% confidence interval. Here, the x-axis is labelled 'Downstream Transferred Transport Distance' as each percentile data ($25^{th}$, $50^{th}$, $75^{th}$ and $95^{th}$) have been shifted / transferred a given distance downstream to obtain the best linear fit with $\ln(IR_{max} - IR_n)$. See text (Sect. 3.2) for description of the method of transferred distance.

lithologies (Figure 8). The application of these theoretical roundness curves to two Himalayan problems is described in Sect. 4.

## 4   Application of Roundness Model to Modern and Ancient Himalayan River Systems

### 4.1   Recycled Modern Pebbles

Today's large rivers of the Himalaya transport pebbles that can be broadly separated into two categories. The first are pebbles sourced from bedrock exposed in the catchment area, and the second are pebbles recycled from conglomerates by ancient river systems (Quick et al., 2019). Because the latter category contains recycled clasts that may have gone through one or more cycles of transport, deposition, and re-entrainment, these pebbles will tend to have greater roundness than pebbles sourced from bedrock exposed in the catchment area.

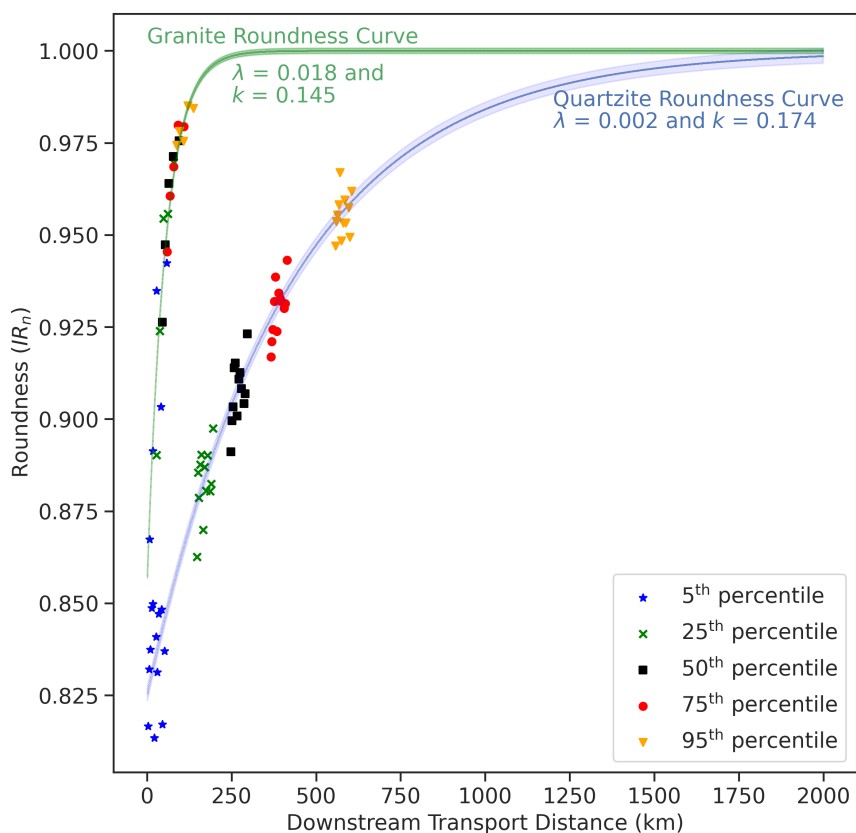

**Figure 8.** Theoretical roundness curve for granite (green) and quartzite (blue) derived from the optimisation method and regression of $ln(1 - IR_n) = f(d)$ field data. Each marker and colour represents field roundness data. The roundness coefficient of granite is nine times that of quartzite. An uncertainty envelope of a 95% confidence interval for both curves is calculated using the standard error of the sample mean.

Quick et al. (2019) studied pebbles in one of the antecedent Himalayan rivers (Karnali River) in the western part of Nepal. They observed more rounded pebbles relative to other Himalayan rivers (Kosi River in eastern Nepal). The difference was attributed to the presence of conglomerate in the Karnali that are not present in the Kosi catchment. The Sub-Himalaya (also know as the Siwaliks) exposes these conglomerates and form the frontal hills north of the foreland basin of the Ganga Plains (See Figure 9). The provenance for the clasts in the Upper Siwaliks conglomerates are meta-sedimentary and metamorphic rocks of Higher and Lesser Himalaya (Zaheer et al., 2022). Consequently, the main channel of the Karnali River consists of first generation pebbles and boulders from the upstream part of the catchment mixed with recycled material from the frontal ranges.

Here, we use field data from Quick (2021). The Sub-Himalaya (Siwaliks) in Karnali region consists of thick (several tens of meters) Miocene-Pliocene conglomerate beds comprising clasts of quartzite, marble, schist, phyllite, dolomite and limestone.

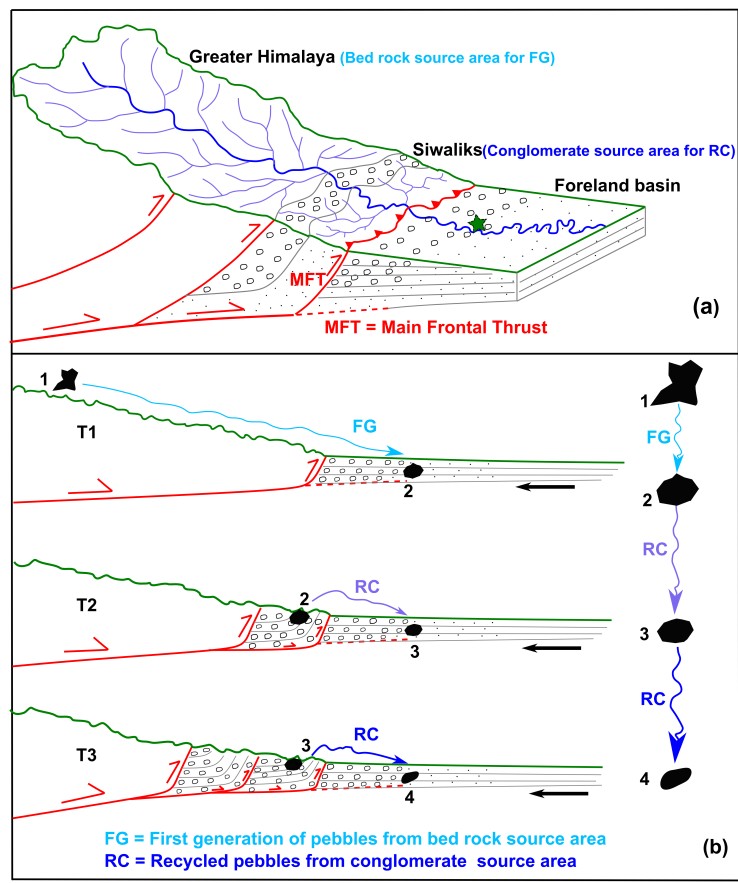

**Figure 9. (a)** Schematic diagram showing a thrust wedge with suggestions for where the major tectonic units of Himalaya would be located relative to each other. This is also representative of the source area for the first generation pebbles (bedrock of the Greater Himalaya) and recycled pebbles (Miocene-Pliocene conglomerate beds of the Sub-Himalaya/Upper Siwaliks) along the Karnali River in western Nepal. The pebble roundness data is taken from Quick (2021), independent of this study. The sample site, marked by a star in the diagram (as well as in Figure 3 (b)), consists of a mixture of modern and recycled sediments. **(b)** Repeated cycles of deposition, uplift, erosion and transportation of sediments in the foreland basin. "T1" represents the deposition time of the first generation of pebbles sourced from the bedrocks of the Lesser Himalaya and Higher Himalaya. "T2" and "T3" represent the times of repeated rounds of erosion of clasts from the conglomerates of the Sub-Himalaya (Siwaliks). The roundness of pebbles increases progressively from 1 to 4. Almost 90% of quartzite pebbles in sampling site (foreland basin) are sourced from the conglomerates of the Siwaliks (Quick et al., 2019). The black arrow represents the direction of convergence of the subducting Indian plate relative to a stable mountain range. The figure assumes the Himalaya have been in a topographic steady state (Thiede et al., 2005), where frontally accreted thrust units are advected into the range during continued convergence resulting in a stable mountain belt width, rather than propagating the deformation front towards the foreland basin.

These clasts from the Sub-Himalaya tributaries are eroded and mixed with the other modern sediment along the Karnali River. The sample site is located in the Indo-Gangetic plain which consists of a full mixture of sediments from Higher to Lesser

Himalaya and Sub-Himalaya (sample site in Figure 9), where $95^{th}$ percentile roundness is 0.995, $50^{th}$ percentile roundness is 0.995 and $5^{th}$ percentile roundness is 0.908 for quartzite pebbles. We compare the modern length of the Karnali River (from channel head to the sampling site) with the transport distance of pebbles calculated using our new model. This calculation is based on the assumption that the quartzite pebbles collected along the Karnali River behave in a similar way as the quartzite pebbles used to generate the roundness curve in this study. The maximum and average transport distance for the quartzite pebbles calculated using the $95^{th}$ and $50^{th}$ percentile roundness is 1472 km and 860 km, respectively (Figure 10), whereas the length of the modern Karnali River from channel-head to the sampling site is only 660 km (see Figure 3 and Figure 9). The transport distance for the pebbles at the sampling site is greater than the length of modern Karnali River.

Using gravel flux calculations and clast analyses, Quick et al. (2019) suggested that quartzite clasts deposited in the foreland basin had experienced at least one round of recycling along the Karnali River. DeCelles et al. (1998) made a similar observation in the Siwaliks sediment further west from the Karnali River, where they found evidence of two rounds of sediment recycling. The interpreted transport distance from our roundness model is greater than the length of the studied modern river, which is consistent with sediment recycling in this setting. In the Himalayan foreland basin, the rapid subsidence of the proximal basin keeps the gravel-sand transition boundary close (10-20 km) to the active front (Dingle et al., 2016). The proximity of the gravel units close to the active deformation front results in them being vulnerable to accretion back into the thrust wedge. Ongoing rock uplift and exhumation of the accreted gravels (now conglomeratic rock) results in renewed fluvial transport of clasts and deposition at the younger gravel units in front of the deformation front (Sinclair, 2011; Sinclair et al., 2018).

The depth of the décollement and the shortening rate control the thickening of the wedge (Dal Zilio et al., 2020) and, consequently, the cycles of accretion and exposure of such gravel stratigraphy. The current shortening rate at the front of the Himalaya is 17-20 mm per year (Bilham et al., 1997; Mugnier et al., 1999; Jouanne et al., 2004). The sediments of the Siwaliks (the source of the recycled pebbles in this study) were diachronously deposited from 14.6 to 1.8 Ma (Mugnier et al., 1999). Assuming constant shortening rates from the time of deposition of the Siwaliks sediment, there has been approximately 250-300 km convergence in last 15 Ma. It is difficult to estimate with precision the extra distance that pebbles could potentially travel through a given number of cycles of recycling in such a context, as the estimate will also depend on the width of the exposure of Siwaliks gravel and on the distance at which these units are exposed upstream of new, emerging mountain fronts; the latter is found to be highly variable (e.g., Quick et al. (2019)). However, this calculation can help us bracket the extra distance that pebbles can travel through one or more cycles of recycling over the last 15 Ma, i.e., tens to a few hundreds of km. The average transport distance of the Karnali River pebbles based on our model is about 860 km, whereas the length of the modern Karnali River from its channel head is only 660 km. This provides a minimum estimate for the recycling distance of 200 km, which is not inconsistent with the calculation of potential recycling distance based on convergence rates. Although these comparisons suggest evidence of recycling, this study cannot determine the number of rounds of recycling or the amount of shortening due primarily to uncertainties in the length of the river channels that drain the Siwaliks, as many of such channels tend to run parallel to the strike of the structures.

### 4.2 Pebbles from the stratigraphic record

A broader application of our method is to calculate the transport distance of ancient river deposits preserved in the stratigraphic record. The Kathmandu Valley in central Nepal is a perched sedimentary basin (Sakai et al., 2006) that has its headwaters south of the main Himalayan drainage divide (Figure 11). There has been some speculation that the location of the headwaters of this catchment were previously in the High Himalaya (Hagen, 1969). In order to test this, we chose to assess transport distances based on rounding of pebbles in Quaternary fluvial deposits.

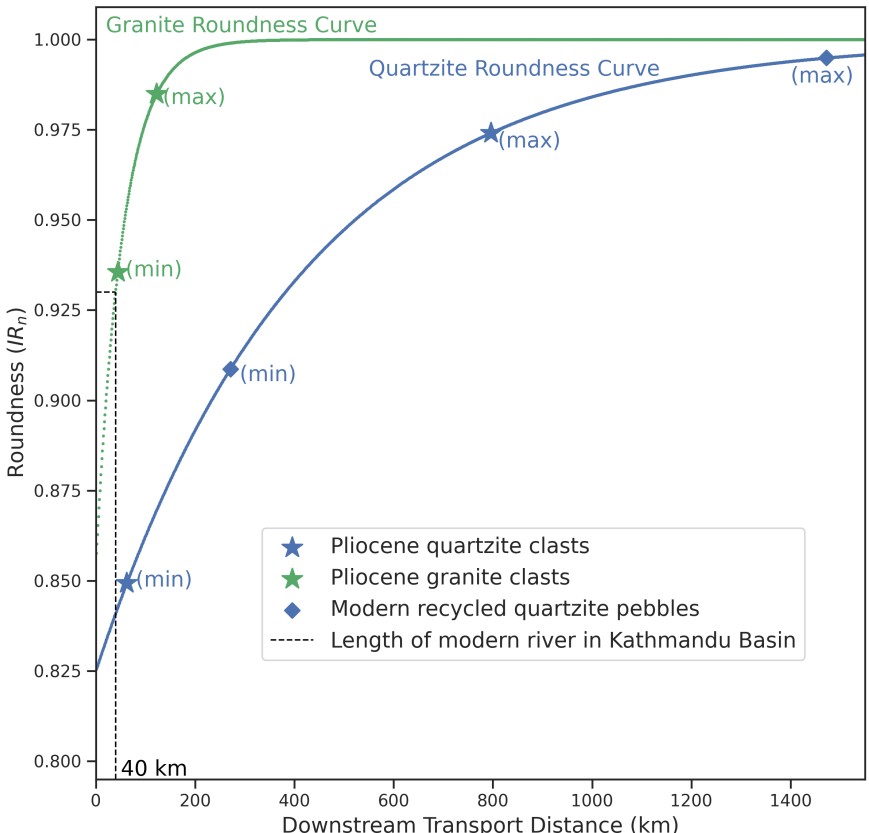

**Figure 10.** Application of our model to Pliocene conglomerate clasts from the Kathmandu Basin and modern recycled pebbles from the Karnali River. The range of likely transport distances for the Pliocene clasts is calculated using the $5^{th}$ and $95^{th}$ percentiles. The minimum calculated transport distance (44 km for granite clasts and 62 km for quartzite clasts) is greater than the length (40 km) of the modern Bagmati River within the Kathmandu Basin. Similarly, the range of likely transport distances for the modern recycled Karnali pebbles calculated using the $5^{th}$ and $95^{th}$ percentiles are 270 km and 1472 km, respectively.

Samples were collected from a site exposed by incision of the Bagmati River (see Figure 3 (b) and Figure 11), which is the main drainage of the Kathmandu Basin. The sampling site is a $\sim 2.5$ Ma old deposit (Yoshida and Igarashi, 1984) that consists

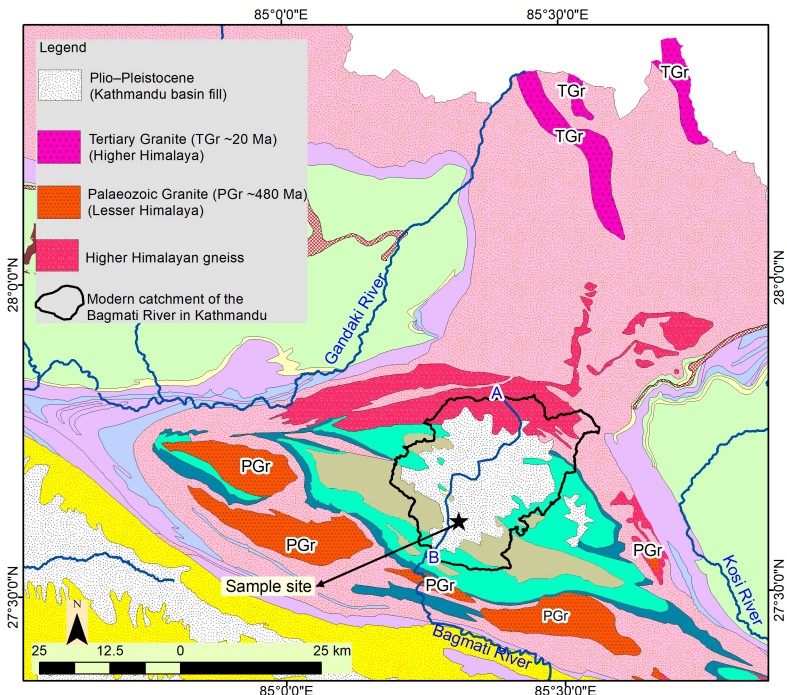

**Figure 11.** Regional geological map showing the occurrence of two possible sources (Tertiary Granite (TGr) in Higher Himalaya and Paleozoic Granite (PGr) in Lesser Himalaya) for the granite pebbles found at the base of the Kathmandu Basin. Note: both granites (TGr and PGr) are outside of the catchment area of the modern Bagmati River in Kathmandu. In the map, A represents the channel head, and B indicates the location of the channel from where the distance along the modern Bagmati River is measured. Map source: Geological map of central Nepal by Department of Mines and Geology in Nepal (Department of Mines and Geology, 2011) and Dhital (2015)

.

of gravel beds underlain by basement rock and overlain by lacustrine deposits (see Figure 12). Quartzite and granite are the most dominant rock types among the clasts found in the conglomerates (marked by star in Figure 12) within the fluvial deposit. It is important to note that the granite clasts are absent in the alluvial fan deposits (top unit in Figure 12). We measured the roundness of both quartzite and granite pebbles to mirror our measurements in modern channels. Based on the $5^{th}$ and the $95^{th}$ percentiles of measured $IR_n$ values, we calculate the range of probable transport distances travelled by the pebbles using

the roundness curve shown in Figure 8. The minimum transport distance (using $5^{th}$ percentile) is 44 km for granite clasts and 62 km for quartzite clasts (Figure 10). The maximum transport distance (using $95^{th}$ percentile) is 123 km for granite clasts and 795 km for quartzite clasts (Figure 10). The measured length of the modern river from channel head (Bagmati River) inside the Kathmandu Basin is only 40 km (Figure 11). The minimum transport distance calculated from the pebble roundness is higher than the length of the modern channel inside the Kathmandu Basin.

In addition, when we investigate the regional geological map (Figure 11) in central Nepal, we do not find any granitic intrusion within the catchment area of the modern Bagmati River inside the Kathmandu Basin. However, there are Paleozoic

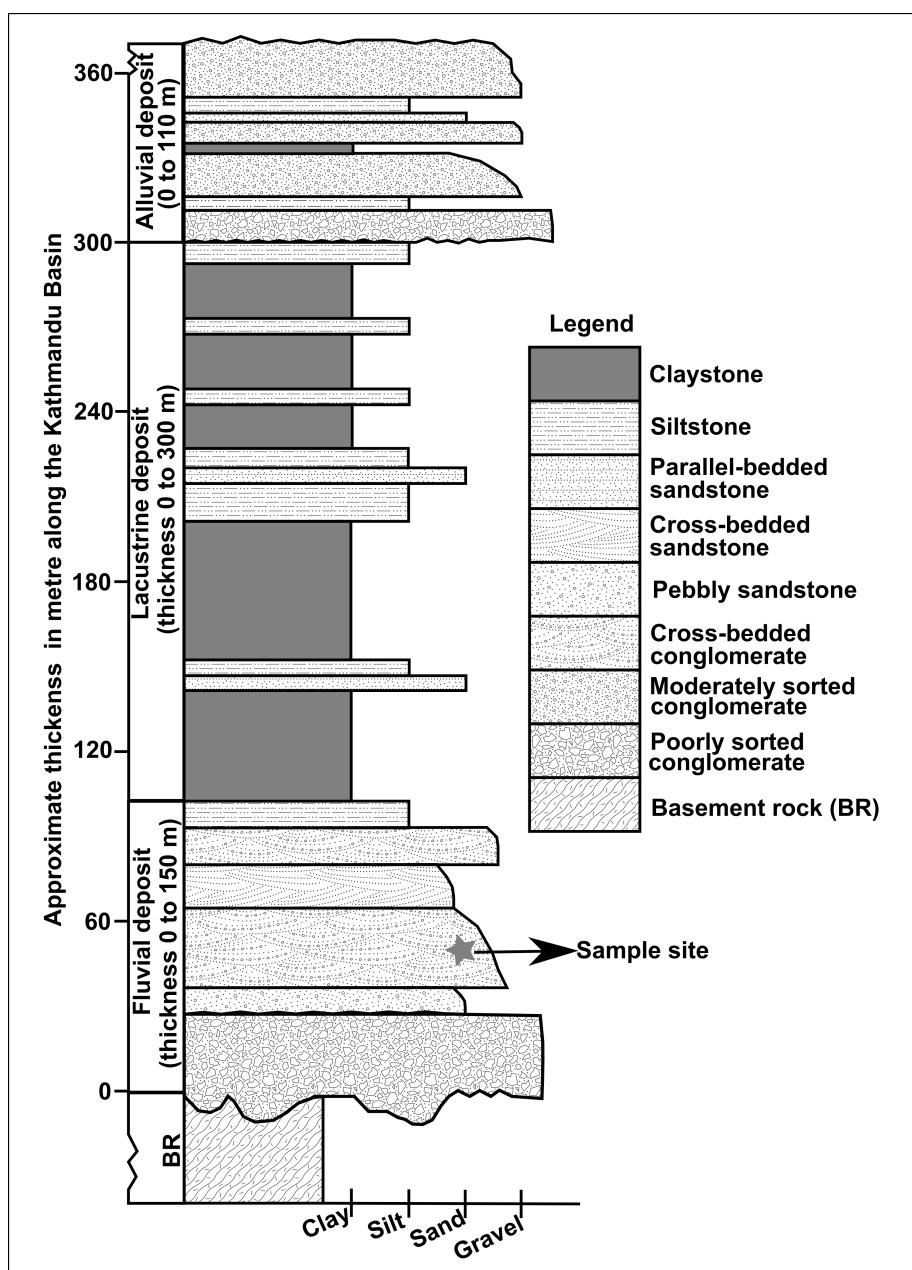

**Figure 12.** A representative sedimentary succession for the valley-fill sediment of the Kathmandu Basin. Pliocene mass-flow type conglomerates and fluvial gravel-sand deposits unconformably overlie the Paleozoic basement rock. Above these deposits, Pliocene-Pleistocene lacustrine clay-silt sediments mostly fill the central part of the basin. Notably, in the southern part of the basin, high-level terraces have formed representing alluvial fans of Pliocene-Pleistocene age.

granites just outside the Kathmandu Basin to the south, and Tertiary granites are located outside the basin to the north (refer to the regional geological map of central Nepal in Figure 11). Consequently, the greater paleo-transport distance of Pliocene granite clasts and the absence of granite source rock in the modern catchment area of the Bagmati River supports the previously hypothesised extensive drainage network (Hagen, 1969) through the present Kathmandu Basin.

While the location of mapped units may have been sufficient to conclude that the granite clasts were sourced from outside the basin, modern geological maps are not always entirely reliable for identifying the source of clasts, particularly in regions where identical lithologies exist in different stratigraphic positions. For example, in Figure 11, the geological map indicates two potential sediment source areas: Paleozoic granite south of the Kathmandu Basin and Tertiary granite north of the basin in the High Himalaya. By relying solely on the geological map, we cannot definitively determine the source area for the granite clasts observed at the base of the Kathmandu Basin. In addition, mapped units may have been exposed in different places 2 Ma ago, as high erosion rates, such as those found in the Himalaya, can result in the removal of kilometres of rocks over millions of years. Here, the high transport distance calculated for granite and quartzite pebbles in the Kathmandu Basin suggests that they were deposited around 2.5 Ma ago by trans-Himalayan rivers, demonstrating the benefits of our roundness model to narrow down the sediment source area for paleo-river channels.

## 5 Discussion

A number of researchers have previously evaluated the roundness of pebbles transported by rivers. For example, Wentworth (1922), Mills (1979), and Miller et al. (2014) discussed the downstream evolution of pebbles along the river system. Wentworth (1923), Mills (1979), Miller et al. (2014) and Gale (2021) suggested a non-linear relationship between roundness and transport distance; here, we have expanded on this by empirically relating roundness to transport distance in the setting of the Himalayan rivers. The model we propose is not limited to the single median or average value for a particular location as done by Wentworth (1923), Quick et al. (2019) and others. The reason why we consider the percentile distribution is that from field observations, we observe that pebbles of identical lithologies, similar size, and the same transport distance downstream may exhibit very different roundness values (see Figure 13). By combining different percentile groups of roundness, we are able to construct rounding curves, using a model that is consistent with a non-linear relationship between rounding and transport distance proposed by previous studies. However, this model comes with uncertainties, some of which are discussed below.

One of the main uncertainties comes from the fact the number of grains measured at each site varies (Figure 5 (a) and (b)). In the case of granite pebbles, we measured an average of 30 individual pebbles at each site. In such cases, the lower and higher percentiles may be represented by a single to a few grains. The maximum number of grains at each site was controlled by availability (as shown in Figure 3 (d); granite is not the predominant rock type exposed in the catchment area), and we accept that this limitation is inherent to our work. We could have discarded percentile values that are driven by a number of clasts smaller by a threshold. However, the choice of a threshold value would have been arbitrary and such an approach would lead to discarding much of the data in the higher ($95^{th}$) and lower ($5^{th}$) percentiles; in addition, all intervals will also be influenced by a few grains. We therefore opt to present and use all the data available. We believe that the changes in the downstream

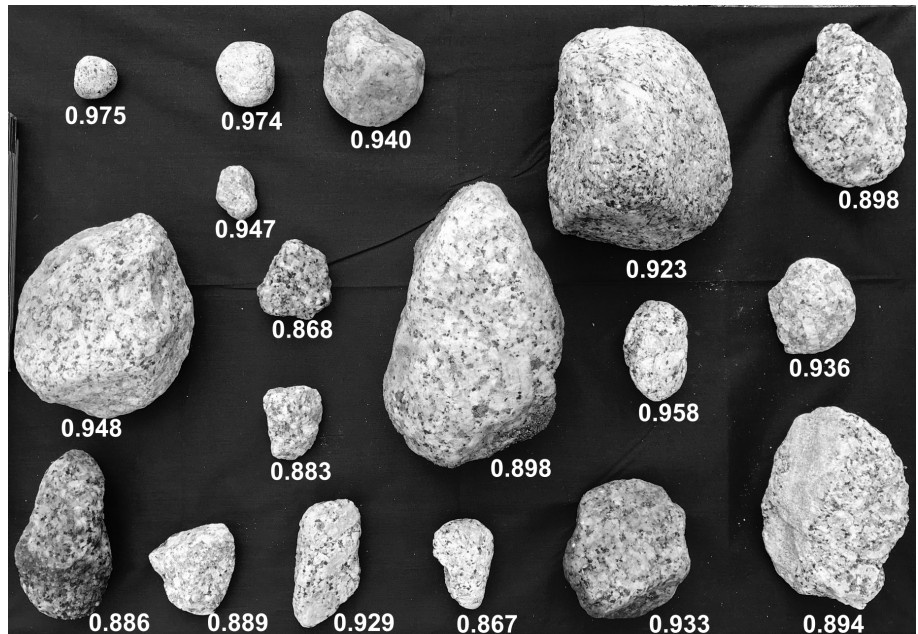

**Figure 13.** Photograph showing the $IR_n$ value of the clasts at location "a" ($\sim$ 8 km downstream from channel head) in Figure 3 (d). Note that the roundness value for this location ranges from 0.867 to 0.975. Although the pebble with $IR_n = 0.975$ has travelled only 8 km from the channel head, its roundness is equivalent to that of pebbles which have travelled 50 km transport distance.

distributions of the roundness data are real (Figure 5 (a)) and hope that this work encourages other researchers to use the same strategy in other locations with better constraints to test the model.

Variations in the coefficient of roundness ($\lambda$) are observed among different rock types. The $\lambda$ for quartzite and granite pebbles from the Himalayan river were calculated using field data, and it was found that granite is significantly more susceptible to rounding than quartzite. This result is consistent with previous studies on the control of lithology on abrasion and roundness (Kuenen, 1956; Sneed and Folk, 1958; Kodama, 1994; Lindsey et al., 2005; Sklar and Dietrich, 2001). Sklar and Dietrich (2001) used an experimental setup to study bedrock erosion rates and processes. They found a relationship between bedrock abrasion rate and the tensile strength of the bedrock slabs used in the experiments. One may wonder if similar relationships may be found between tensile strength and the rounding of pebbles (which is driven by abrasion). Based on the rock descriptions provided, the granite and quartzite pebbles used in this study are equivalent to the weathered granite (#16) and quartzite (#28) of Sklar and Dietrich (2001).

According to the relationship illustrated by Sklar and Dietrich (2001), the tensile strength of quartzite is almost six times greater than that of weathered granite, which mirrors the greater roundness coefficient of granite, almost nine times greater than that of quartzite, observed in our work. Similarly, Sklar and Dietrich (2001) used two types of granite: a weathered granite (#16) and a granite (#27). They found that the tensile strength of granite is several times greater than that of weathered granite. This indicates that tensile strength varies even within the same rock types, and that the rounding coefficients will likely reflect

these differences. The two parameters of our roundness model (prefactor and roundness coefficient) are therefore likely to be dependent on bedrock lithology, but also tectonics and climate that influence weathering type and rate, and dominant hillslope sediment processes. These factors will determine the distribution of initial grain size and abrasion processes, as suggested by Sklar and Dietrich (2001).

The prefactor ($k$) in the model indicates the predicted starting roundness of pebbles at a distance ($d$) = 0 km. However, the roundness of pebbles, even of the same rock type, varies significantly at a particular location. For example, in Figure 13, $IR_n$ ranges from 0.867 to 0.975 for granite pebbles from the most upstream location "a" along the Rapti River in our study area (located in Figure 3 (d)). Although the pebble with a roundness value $IR_n = 0.975$ in the top left corner of Figure 13 had travelled only 8 km from the channel head, it possesses a roundness value equivalent to that of pebbles that had travelled 50

480 km (see Figure 5 (a)). Clearly, clasts fed into a river, even from areas of the same rock type with similar sediment production processes (weathering, erosion, and transportation), will not have identical roundness. The different mechanism of sediment production (such as landslide, hillslope erosion), exposure to weathering conditions, and boundary conditions during the flow (even in the first km of transport), may impact the roundness of pebbles. Both experimental observations and modelling results indicate that sediment undergoes intermittent motion, resembling a succession of periods involving "flight" and periods of

485 rest, as described by Lajeunesse et al. (2010). Additionally, we hypothesise that some pebbles that travel through cavities and pools (potholes) along the upper reaches of the river channel may become more rounded despite shorter -along stream-transport distances. This implies that each pebble may have a different rounding history based on its initial roundness, transport mechanism, and history of transport. Our model assumes that all pebbles experience the same rounding history, which may be overly simplistic and raises questions regarding how round pebbles can be after a given apparent transport distance.

In the Nepal Himalaya, the length of the longest river from the channel head to the gravel-sand transition is approximately 1000 km, and many other rivers in the world have a length exceeding 1000 km. In general, most pebbles will not survive a transport distance of thousands of km; however, quartzite pebbles are extremely resistant and can survive transport distances of thousands of km (Dingle et al., 2017). Attal and Lavé (2009) document abrasion rates as low as 0.1% mass loss per km for quartzite pebbles. At this rate, a quartzite pebble will lose 86% of its mass in 2000 km. Sediment recycling can allow

pebbles to cover additional distances. For example, the maximum sediment transport distance for recycled pebbles along the Karnali River (location marked with a star in Figure 3 (b)), calculated using our model, is approximately 1500 km. Although we calculated the maximum transport distance using the $95^{th}$ percentile value, there are some other pebbles in the population which are perfectly rounded (roundness value ~1.0). According to our model, a distance of approximately 2000 km is where quartzite pebbles achieve perfect roundness (Figure 8).

We emphasise that the prefactor ($k$) and roundness coefficient ($\lambda$) derived from our study are specific to quartzite and granite pebbles found in the two small catchments of the Himalaya. The quartzite pebbles are sourced from a massive bed of monomineralic (quartz) quartzite with no or slight degree of weathering. The granite pebbles are rich in feldspar and mica minerals, making them susceptible to rapid weathering and abrasion. Conversely, granite pebbles from another region may be less rich in feldspar and mica, rendering them more resistant to weathering and abrasion. Consequently, variations in

mineral composition may imply different pre-factor and roundness coefficient values compared to those proposed in this paper.

Therefore, the values obtained in this study should not be used as universal values for all granite and quartzite pebbles. However, we do believe that the non-linear relationship between roundness and transport distance proposed here may be applicable to all pebbles where downstream rounding is controlled by abrasion. It is important to note that pebbles sourced from thinly bedded (shale) or highly foliated (slate, schist) bedrock may have a different story when it comes to downstream rounding, where the downstream changes in shape and size are driven by processes such as fracturing. However, it is noteworthy that the parameter ($IR_n$) used in this model to represent the roundness is independent of sphericity. Thus, while we acknowledge that the roundness coefficient and prefactor may vary in other catchments based on factors such as hardness, lithology, climate, tectonics, and sediment production process, we believe that the concept of the model will still be valid.

We apply our new roundness model to ancient and modern sediments. Using the roundness curves generated from this study, we estimated transport distance for the pebbles. Nonetheless, uncertainties associated with the estimated transport distances are significant. We use roundness data measured by Quick (2021) for the modern pebbles along the Karnali River. The roundness for the Pliocene clasts of the Kathmandu Basin is measured in this study. The estimated transport distance assumes that the pebbles / rock fragments along the Karnali and the Pliocene Bagmati River had the similar initial roundness as used in this model. Additionally, the Karnali River and the Pliocene equivalent of the Bagmati River are trans-Himalayan rivers, while the roundness coefficient ($\lambda$) and prefactor ($k$) used in this study are derived from smaller Lesser Himalayan catchments. Consequently, the estimated transport distance assumes similar abrasion processes/rate in both trans-Himalayan rivers and smaller rivers with limited catchment area, which is a strong assumption. Furthermore, glacial sources for trans-Himalayan rivers are likely to provide fragments with different roundness values on entering the fluvial system.

One may question whether rounding is dependent on the size of the grains considered. In this study, pebbles ranging from granules to cobbles were collected from the river's channel deposit. We did not categorise the sediment based on grain size, as Quick (2021) showed no correlation between downstream fining and rounding of grains. Whether our model is applicable to downstream rounding of fine particles, such as sand and silt, is uncertain because it depends on the nature of the abrasion processes and hence grain interactions during transport. Naturally, the more sediment is transported in suspension, the less it will experience abrasion.

Overall, we present roundness curves based on field-measured data, which we believe are applicable to most grains experiencing rounding in fluvial environments. The parameters ($\lambda$) and prefactor ($k$) encompass information related to various factors, and we believe future research will isolate the specific impacts of these factors. Most notably, this study establishes the groundwork for quantifying transport distances based on sediment roundness. We advise researchers who want to use our model to estimate pebble transport distances to identify lithologies that are unique within a given catchment, and to measure the roundness distribution of pebbles of that lithology (several pebbles, ideally at least 100) at specific locations downstream. The smaller the distance between sampling sites and the greater the distance covered, the better the rounding curves will be constrained. Subsequently, users can determine the percentile roundness that best represents the bulk of the sediment at each location to approximate the transport distance. This decision may involve a degree of subjectivity on the part of the users. The uncertainty in the measurement lies in the range between the transport distance calculated using the minimum and maximum percentiles, indicating the minimum and maximum possible transport distances, respectively. Selecting a location in the

main channel, where sediment is well-mixed and away from lateral tributaries originating from nearby distances, and distant from areas where sediment is transported laterally from hillslopes, landslides or human activities, will improve the overall estimation.

## 6 Conclusions

A workflow that measures pebble silhouette using a colour threshold to differentiate the background and pebble area in 2D photographs has been proposed that enables automated extraction of pebble roundness and hence ensures the replication of measurements. The normalized isoperimetric ratio ($IR_n$) is calculated based on pebble outlines and is used to parameterise the roundness (/angularity) of modern river pebbles and ancient clasts in the stratigraphic record. This parameter is easy to measure and isolates roundness from sphericity (unlike the isoperimetric ratio). The method has been applied to pebbles from

two rivers that drain the Lesser Himalaya and frontal regions of the Himalaya, which have either granite or quartzite exposed in their headwaters. Consistent with previous studies, we show a nonlinear relationship between transport distance and pebble roundness. We propose a new model that mirrors Sternberg's law of abrasion. Using field data, we generate roundness curves for two lithologies (quartzite and granite). These curves provide estimate of transport distance beyond our study reaches. Using our new model, we demonstrate that our field data are consistent with rates of rounding that decrease exponentially

with distance downstream. The degree of rounding as a function of travelled distance downstream varies as a function of lithology, with granite pebbles rounding nine times faster than quartzite pebbles. Using the calibrated rounding model, we give support to previous studies that indicate that the recycling of clasts from conglomerates impacts the degree of rounding of some trans-Himalayan rivers as they exit the mountain front. Conglomerates from Pliocene fluvial deposits in the Kathmandu Basin comprise rounded quartzite pebbles that suggest paleo-transport distances that are greater than the current length of the

Bagmati River in the basin. We propose that this supports previous suggestions that the paleo-Bagmati River was significantly larger than the present-day equivalent, and that the catchment has reduced in size due to interactions between tectonics and erosion leading to drainage reorganisation.

*Code and data availability.*  The code and data are stored in the following data repository: doi.org/10.7488/ds/7515

*Author contributions.*  Funding and project design was by HDS. MA and PP developed the methodologies. PP conducted the fieldwork and 565 data analysis. MA, SMM and MN provided input on data analysis. PP wrote the paper. All authors contributed to revising the text and figures.

*Competing interests.*  At least one of the (co-)authors is a member of the editorial board of Earth Surface Dynamics.

*Acknowledgements.* We express our gratitude to Subash Acharya, Puspa Raj Dahal, Mahesh Raut, Ravi Nepal, and Khim Bahadur Khadka for their invaluable support during the fieldwork. Additionally, we extend our thanks to the Tomorrow's Cities' Kathmandu Hub for generously providing financial support for this research. We also acknowledge that this study forms a part of a Ph.D. program funded by the School of GeoSciences-University of Edinburgh (UKRI GCRF Tomorrow's Cities Hub, Grant number: NE/S009000/1).

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
