# Peer review of "Downstream rounding rate of pebbles in the Himalaya"

_EGUsphere, 2023_

## Referee Comment (RC1)

This is my review of the manuscript entitled "Downstream rounding rate of pebbles in the Himalaya" submitted by Pokhrel et al for publication in Earth Surface Dynamics. In this work, the authors propose a relationship between the the roundness of coarse sediments found in rivers and transport distance. The manuscript is well written and presented, and supported by adequate and readable figures. The idea is not new but the authors developed a new formalism that allow for quantification. They show how this model can be used to reconstruct transport distances of old deposits, leading to discussion about sediment recycling and drainage reorganisation. Overall, I think this is a great work with large potential for the community that should be published. However, I think that the current manuscript requires some extra work before publication. First, I feel that the manuscript needs some reorganisation (for example, some methods appear as Results, or Discussion is rather a conclusion). Second, I missed some discussions about the implications of this work. Please find below my detailed comments that will hopefully help. I have no doubt the authors can address my major concerns so I'm looking forward to reading this manuscript in Esurf.

Laure Guerit (CNRS, Géosciences Rennes, France)

**Introduction**

I found these pages interesting, however, they look a bit like a report rather than a paper introduction. In the current form, it is quite difficult to identify what is already known,/done how it is used and what are the current flaws that motivate this study. I strongly recommend to shorten this part in order to be more focus and somehow more conclusive about what is known and not known on the relationship between roundness and transport distance. This would really highlight the relevancy of this work.
Grain size is also used to discuss transport dynamics so it could be interested to discuss briefly what can be done with it and why shape is more interesting.

**Materials and Methods**

From section 2.1, I didn't really understand if you use a metric that has been developed by others (as suggested by the various references) or if you add something new (as suggested by line 159). In addition, I didn't understand the motivations to use this definition of roundness more than another. I think this could be clarified in the revised introduction. Similarly, if I understood correctly, you didn't develop new algorithm for image processing but you used options that are already available in ImageJ. It is totally fine but it is not that clear from the current section 2.4. Also, there are several methods available to segment grains and extract geometrical information (see for example BaseGrain, PebblesCount, G3point) so this could be explored a bit in the introduction. Some might avoid the use of 2 softwares including one commercial (ImageJ and ArcGIS) and make your methodology more open access.

In addition, I feel that the level of details is sometimes too high (for example, I don't think you need to mention that you used HCl to identify rocks - again, this gives the feeling of a report rather than a paper). There are also some repetitions (for example, l. 208 is a repetition of L. 184, l. 211 is a repetition of l. 193).

**Results**

As a general comment, a lot of this section sounds like methods and should be presented as such, or at least in a part dedicated to "Modelling" for example.
I like your interpretation that the percentiles of the roundness distributions are families of pebbles evolving along the river. However, I'm a bit concerned by the limited number of points in the lower quantiles. With on average 30 samples for the granite, the 5th percentile might be represented by 1 grain, a few at best. I would recommend to have a minimum number of samples (maybe 5?) in a percentile before to use it.
I was a bit confused at first by the statement line 287 as panel c on Figure 4 shows that angular grains round faster than the others. I assume you mean that whatever the initial angularity, once two grains have the same roundness, they continue to round at the same rate. I suggest to rephrase a bit this part to make it more clear.

A major issue of this part is that I didn't find a definition for the transferred distance. I assume that it is the total transport distance of the grains, which is different from the distance to the headwaters to due drainage reorganisation and recycling. Please add a clear sentence about it (and I think it should be defined in the Methods, not in the Results).

On Figure 7, you show the theoretical rounding curves for the two populations of grains. I really like it but I have a few questions:
 - do you think it is possible to derive an enveloppe, rather than a curve, in order to estimate uncertainties ?
 - The x-axis goes up to 2000 km, yet, it is quite unlikely that pebbles survive that long (see for example Dingle et al, 2017). Therefore, what was the motivation for such a long x-axis and in which configuration do you expect pebbles to live that far?

**Section 4**

I really like this part, but I would have appreciate to see it as Results followed by a proper discussion. I understand the potential impact of recycling on total transport distance, however, the way it is shown in Figure 8 will not increase the distance. In order for recycling to have a significant impact of total distance with respect to the position where the grain is found, you need to have shortening. This leads to two comments:
 - The results should be discussed with respect to the known shortening rates in the area.
 - Do you think your approach could be used to estimate shortening from the difference between position and total transport distance?

It is interesting to compare with the geological map, but I think you could better articulate the two approaches. Somehow, the map confirms your findings but it would have been easier to simply look at the map and identify that the grains must come from an area that is no longer part of the catchment. I think you could explain a it more the advantages of your approach in this kind of context (this comment is related to my first comment on the Introduction).

**Discussion**

This section is more a long conclusion. It's fine but in consequence, I missed a proper discussion on the results: is there a relationship between the initial size and the initial angularity? How can rock breaking affect the rounding curve? More specific to the results on the Himalaya: are they evidences for major drainage reorganization as suggested by the transport distance? What about the transferred distances? How can we use this approach elsewhere? It would really strengthen the manuscript.

**Conclusions**

Please make a it more clear what is new results from this study and what is not.

**Minor Comments**

- l. 14 "eight times that of quartzite" : it is written "seven" in the caption of Figure 7. Please correct.
- l. 66: Feher et al is a preprint that has been withdrawn by the authors. It can't be used as a reference.
- l. 191 missing ( before "Mudd"
- Lines 237 to 266 should be in Appendix rather than in the main text.
- Fig. 4 The text at the top of each panel is quite close to the boxes. If possible, consider adding a bit of space. Missing captions about the colors on panels a and b (it does not seem to carry any information but maybe it can be use also on Figure 2 ?)
- l. 371 extra capital s "Sample"
- Fig. 9 missing "a" in quartzite
- Fig. 10 what are A and B for? Missing space in the caption between Basin and Note.
- l. 409 extra "

---

## Referee Comment (RC2)

Review of the manuscript "Downstream rounding rate of pebbles in the Himalaya" by Pokhrel, et al., submitted to Earth Surface Dynamics

**General Comments:**

The authors present the paper "Downstream rounding rate of pebbles in the Himalaya", which outlines a method for measuring roundness of pebbles, as well as proposes a model for relating pebble roundness to transport distance. They specifically address challenges with previous methods of measuring pebble shape parameters, and identify a method for automatic and repeatable extraction of these shape parameters (namely isoperimetric ratio) from 2D photographs of pebbles using publicly available software. This shape characterization method was applied to pebbles collected from two rivers in Nepal, as well as clasts found within conglomerate deposits in a similar region, which were then used to calculate rounding curves for two rock types within these watersheds. By determining transport distance of historical clasts, the authors drew conclusions about the length of paleo rivers in the Himalayas.

This work contributes a useful approach for relating clast shape to transport distance, especially for river systems where data collection is limited to shorter distances or a smaller number of field sites, or for applications in paleoenvironment reconstruction. Additionally, from my exposure to the pebble rounding literature, it seems that others emphasize rounding as a function of mass loss in order to draw general conclusions, while this study seems more practical for direct use in specific watersheds. However, I do think that this work could be even more impactful if the authors connected their results to the larger context of universal pebble rounding behavior, such that their work could readily be applied to other geomorphic settings. Given discussion of the results within this context, as well as editing for clarity and conciseness, I think that this paper suitable for publication in ESURF and believe that it contributes to the pebble rounding literature. I hope the authors find my comments to be helpful and constructive, and I wish them the best in their research endeavors.

**Specific Comments:**

Broadly, I think that the organization of the introduction (Section 1) can be improved and streamlined. Section 1.1 includes discussion on both previous pebble abrasion research, as well as shape parameters that have traditionally been used to quantify pebble roundness. Subsequently, Section 1.2 discusses shape parameters in depth, then Section 1.3 discusses controls on pebble shape and summarizes previous research in more depth. I would recommend combining the first two paragraphs of Section 1.1 with Section 1.3 and presenting this information first as background on pebble abrasion processes. I would then combine the last paragraph of Section 1.1 with Section 1.2 and present this information as background on shape indices.

Section 1.1, paragraphs 1 and 2 address previous research on pebble chipping/abrasion/attrition. It seems worthwhile to include a sentence or two at the outset on the definitions used in this paper, since previous studies use different terms to refer to specific breakdown mechanisms. For example, in paragraph 2, the authors write that "processes like sandblasting, chipping, and granular removal by crushing or grinding" fall under abrasion and

increase pebble roundness, but then in the same sentence, use the phrase "chipping of large fragments" as a process that reduce pebble roundness. As a reader, I am slightly confused as to how chipping is defined such that is can both increase and reduce roundness, especially since chipping appears to primarily be used in the literature to describe the process of pebble rounding due to bedload transport (e.g., Novak-Szabo, et al., 2018). Additionally, I am aware that crushing or grinding tends to fall under the purview of "comminution", which primarily breaks rocks down into smaller pieces and may increase sphericity, but would not necessarily increase roundness. From my experience with the pebble abrasion literature, the terms "abrasion" and "chipping" tend to be used to describe the small-scale breaking off of edges and corners that progressively round rocks (e.g., Miller, et al., 2014; Szabo, et al., 2015; Novak-Szabo, et al., 2018), while "attrition" is a more general term that could describe small or large scale breakdown (e.g., Miller and Jerolmack, 2021), and "fragmentation" is used for significant breakdown into large pieces (e.g., Novak-Szabo, et al., 2018).

In general, I found Section 1.1 (last paragraph) and Section 1.2 to be rather lengthy in the description of shape parameters used in the literature and methods for calculating those shape parameters. For example, I think that the discussion of automated image processing methods for grain shape detection can be reduced to only methods utilized or built upon by this study. It may also help readers if you state that circularity and isoperimetric ratio are equivalent shape indices earlier in the paper, since some readers may be more familiar with the term circularity.

Section 1.3 addresses the effect of lithology on pebble rounding. I think this section could be strengthened with the addition of background on the relationship between material strength and attrition. Since rock strength is known to control rate of attrition (Sklar & Dietrich, 2001; Wang, et al., 2011 – Abrasion of Yardangs) and has implications for attrition mechanism (specifically for abrasion/chipping in Miller & Jerolmack, 2021), I feel that this may be useful in interpreting rounding of your granite vs quartzite pebbles.

I think that the inclusion of a figure (or addition to a pre-existing figure) could improve the discussion in Section 2.1 regarding selection of the normalized isoperimetric ratio as the roundness parameter. The reader might be able to better conceptualize the isoperimetric ratio and normalized isoperimetric ratio for different pebble shapes if there was a figure showing shapes and their *IR* values. Figure 12 already does this for normalized isoperimetric ratio.

In Figure 2 panel b, it may help to draw the reader's eye to the location of the study catchments by outlining them in red or bolding the catchment names. Currently, the brown outline is a similar color to mid-elevations on the map.

The last three paragraphs of Section 2.4 include specific instructions for pebble shape extraction and measurement in ImageJ and ArcGIS. I feel that the explicit step-by-step instructions are unnecessary and could largely be eliminated, or included as an appendix, to shorten the paper.

Sternberg's law is first brought up in Section 3.1 (other than the abstract). Given that this is the basis of the proposed abrasion model, it may be appropriate to introduce Sternberg's law in the introduction section of the paper.

In Figure 4, you may want to increase font or bold the panel labels (a., b., c., d.) since they are a similar font size to the letters used for site locations.

The Figure 8 identifies a sample site along the Karnali River. When the figure is first referenced in Section 4.1, it is not immediately apparent to me whether additional field data was collected during this study, or of the same site is from Quick, et al. (2019). Later in Section 4.1, you state that the recycled pebble field data is from Quick, et al. (2019), but it may help to clarify this in the figure caption. Further, in Section 4.2, you mention more field data from the Bagmati River collected during this study. It may be relevant to briefly introduce this field site in the methods section, as well as place this site and the Quick, et al. (2019) site on the map in Figure 2.

In the discussion section, the authors address the coefficient of roundness, $\lambda$, and prefactor, $k$, which corresponds to initial shape of the pebbles. Since $\lambda$ varies for the two rock types in this study, it would be interesting to address the relationship between material strength and the coefficient of roundness. There is also discussion of how the prefactor, $k$, varies depending on a variety of factors. Domokos, et al. (2015 – Universality of fragment shapes) show that fragmented rocks have a general mass and shape distribution. Assuming that rock fragments entering upland river systems in the Himalayas were generated by energetic processes, it might be interesting to address how the prefactor could be generalized across different watersheds. Further, assuming a general size distribution for initial fragments and that bedload dominates subsequent transport, a universal mass loss curve for particle rounding by bedload can be reached (Novak-Szabo, 2018). While this universal rounding curve relies on knowing the mass of an initial particle, perhaps you can discuss how universal behaviors, along with material-specific properties, can allow you to generalize your model and determine transport distance across a variety of watersheds/conditions.

Additionally, there is discussion of rapid rounding of granite pebbles within 8km of the source. I am interested whether the authors considered utilizing shorter distances, rather than 50km for the distance over which to fit a linear regression. In Figure 5a., the median roundness values appear to show the expected relationship where roundness increases toward 1 over the surveyed distance. Additionally, other studies (Miller, et al., 2014; Novak-Szabo, et al., 2018) observe the expected rounding curve over distances of ~10km in the field. Since some granite pebbles are already fairly rounded at 8km in your study area, I expect that even this short distance is sufficient for noticeable shape changes to occur. Would the results change if the same analysis was applied over shorter distances? Would the results agree if the authors compared them to a more traditional rounding curve applied to the 50km over which the granite clasts were collected?

**Technical Comments:**

L14: In the abstract, you state that the roundness coefficient is 8x greater for granite pebbles, but later state that it is 7x greater.

L23: Recommend use of semicolon or em dash rather than colon as punctuation after "This also applies to modern rivers".

L26: Same as above regarding the colon after "not limited to Earth".

L115: Appears to be a typo; should read "how *the* pebbles round".

L175: Might consider referencing Figure 2c at the end of the sentence.

L181-182: Sentence is slightly confusing; I would recommend indicating that there is both quartzite and granite in *upstream reaches*, but quartzite bands are exposed downstream.

L188: No need for comma in this sentence.

L191: Citation for Mudd et al., 2022 should be in parentheses.

Figure 2 caption: The word "lithology" is misspelled in Line 3.

L217: Need a comma before the word "but".

L240: Recommend use of semicolon rather than colon as punctuation after "object in a raster environment".

L271-272: Consider rewording sentence for clarity.

L278: Recommend use of semicolon or period instead of colon.

L330: End quote of 'scipy.optimise.minimise' is facing outward.

L331: Start quote of 'Nelder-Mead' is facing outward.

L333: Start quote of 'Nelder-Mead' is facing outward.

L336: End quote of 'minimise' is facing outward.

L342: Recommend use of semicolon rather than colon for punctuation.

L342-343: Granite rounding coefficient is 7x that of quartzite differs from abstract.

Figure 7 caption: Rounding coefficient differs from abstract.

L358: Should be "boulders" rather than "boulder".

L360-362: Sentence is confusing to follow.

L364-365: I also find this sentence somewhat confusing.

L371, 372: The letter "s" in sample should be lower case.

L377: Should be "behave".

L379, 380: Missing word – *the* sampling site.

L391: Missing be comma before the word "which".

L398: The word clast should be plural after granite and quartzite.

L401: Missing word – *the* modern channel.

Figure 9 caption: Missing *the* before "modern river"; could also say "modern Bagmati river" for succinctness.

Figure 10 caption: Needs space after period in Line 2.

L403-404: The Kathmandu Basin is repeated twice in the same sentence.

L408: End quotes at the end of the sentence are unnecessary.

L411: Word pebble should be plural.

L417: Missing end parenthesis after Figure 12.

L435: Citation for Lajeunesse, et al. 2010 should not be in parentheses.

Figure 12 caption: start quote before 'a' is facing out.

L456: Missing word – *a/this* new roundness model.

L456: Extra *the* before "ancient and modern sediments".

L467: "fluvial environment" should be plural.

Section 6: Recommend paragraph form rather than bullet points.

L471-472: Extra *the* before word "pebble" and "2D".

L478: Should remove "and" and replace with a comma.

L480-482: Run-on sentence; recommend rephrasing.

L480: Start quote for 'Sternberg's Law' is facing out.

---

## Community Comment (CC1)

Review of the manuscript: "Downstream rounding rate of pebbles in the Himalaya"

Submitted to: Earth Surface Dynamics

Reviewed by: Joakim Edlund

The study "Downstream rounding rate of pebbles in the Himalaya" by Pokhrel et al. presents a new method to quantitatively determine the fluvial transport distance of pebbles by clast roundness. Two modern rivers in the Himalayas of Nepal are used to develop the method which is then applied to recycled pebbles from the Karnali River and sedimentary deposits in the Kathmandu Basin.

I found the paper to be a very interesting and novel read. The introduction section gave a great overview of the challenges and developments in grain shape characterization. I highly appreciate that the methods are described with a great deal of detail, making it relatively easy for other researchers to replicate the method. The figures presented in the study are phenomenal, greatly aiding the reader's understanding of the results and concepts of the study. The methodology of the study may be applied to many other catchments and sedimentary deposits around the world to gain insight into the geomorphic history of the area. The study forms a groundwork for assessing transport distances based on clast roundness which may serve as a baseline for future research to build upon.

I would recommend accepting the paper after a few corrections have been made based on the following minor comments.

**General comments:**

Consider using larger fonts for headings and subheadings. Personally, it helps me find where sections begin and end. The main headings in particular should be bigger to stand out from the rest of the text.

In section 4.1, it is stated (on lines 362-364) that the number of cycles of uplift, erosion and re-deposition of conglomerate pebbles can be addressed using the model, but unless I am missing something, I can't see this discussed anywhere else in the paper. Evidence for recycling of pebbles in the Karnali River is provided later in the section, stating that some pebbles in the modern river had been rounded to an equivalent of 1472 km of transport despite the length of the river only being 660 km. This would then imply that recycling was responsible for the other ~800 km equivalent of abrasion. However, nowhere is this result translated into an estimate of how many cycles of uplift, erosion and re-deposition have occurred. Furthermore, it is not stated how the model could be used to provide such an estimate.

In my opinion, the placement of figures is sometimes a bit odd. Figure 1 is mentioned one time in the text, this being at the top of page 4, but the figure itself is shown at the bottom of page 5, far away from the mention in-text. Figures 10 and 11 share a similar issue, but more egregiously so. In this case, the first in-text mentions of figures 10 and 11 are found near the bottom of page 19, but the figures appear much later. Figure 10 is shown at the top of page 21 while figure 11 is all the way down on page 22. Since figure 10 is mentioned many times above and below the figure it is not a significant problem, however, figure 11 is only mentioned on page 19 and nowhere else, making its placement several pages below (in the Discussion section rather than Results) rather confusing. The issue with having figures placed far from their in-text mentions is that it forces the reader to scroll back and forth through the document while reading, sometimes over several pages, which is detrimental to the reading experience. Of course, a perfect layout is not always possible, but in my opinion the article would be improved if the back-and-forth scrolling could be reduced, perhaps by making some figures smaller and/or moving them around.

**Line comments:**

L13-14: In the abstract it is mentioned that the roundness coefficient for granite pebbles is eight times that of quartz pebbles, however, it is claimed on lines 342-343 and in the figure 7 description that the roundness coefficient for granite pebbles is seven times that of quartz pebbles. Whichever of these two stated numbers is more accurate should be used consistently throughout the paper.

L115: "…will likely influence how they pebbles round" Typo, 'they' should be 'the'.

L119: Informal language, usage of "don't" instead of "do not".

L151: I'm unsure why the spacing above and below equation 1 is uneven. Equation 4 clearly has even spacing, while I can't tell for sure with equations 2 and 3.

L175: Unnecessary space before the '~' in "( ~100 m)".

L220: Missing comma after "For example".

L280-281 & L283: Almost the exact same sentence ("Granite pebbles are rounder than quartzite ones when comparing the percentiles across lithologies") is repeated twice. Was the first instance of the sentence supposed to be removed?

L349: Given the context, the word 'latter' should be used instead of 'later'.

L360-362: Consider rewording this sentence to make it easier to follow.

L367: There is a space before the '.'.

L368: "…in this region consists thick…" I believe "consists of" would be correct here.

L408: The line ends with a quotation mark, likely a typo.

L435: Given the wording of the sentence, the reference should not be in brackets but rather part of the text.

L445: There is an extra space before the closing bracket in "($IR_n$ )".

L456: The sentence "We present the applicability of new roundness model to the ancient and modern sediments" doesn't quite work. Did you mean to say "…of a new…"?

I hope my comments prove helpful and I wish you the best of luck on your research.

---

## Author Comment (AC1)

**Authors' Response to Reviews of**

**Downstream rounding rate of pebbles in the Himalaya**

P. Pokhrel, M. Attal, H. D. Sinclair, S. M. Mudd and M. Naylor EGUsphere Preprint, https://doi.org/10.5194/egusphere-2023-2157

RC: Reviewers' Comment, AR: Authors' Response,

Manuscript Text

**1. CC1: Joakim Edlund**

AR: Dear Joakim Edlund,

Thank you very much for expressing interest in our work and for your appreciation, as well as for providing valuable comments and suggestions. Your insights have significantly contributed to the enhancement of our manuscript, and we sincerely appreciate the time you dedicated to this manuscript.

Kind regards, Pokhrel et al.

- RC: The study "Downstream rounding rate of pebbles in the Himalaya" by Pokhrel et al. presents a new method to quantitatively determine the fluvial transport distance of pebbles by clast roundness. Two modern rivers in the Himalayas of Nepal are used to develop the method which is then applied to recycled pebbles from the Karnali River and sedimentary deposits in the Kathmandu Basin.
- RC: I found the paper to be a very interesting and novel read. The introduction section gave a great overview of the challenges and developments in grain shape characterization. I highly appreciate that the methods are described with a great deal of detail, making it relatively easy for other researchers to replicate the method. The figures presented in the study are phenomenal, greatly aiding the reader's understanding of the results and concepts of the study. The methodology of the study may be applied to many other catchments and sedimentary deposits around the world to gain insight into the geomorphic history of the area. The study forms a groundwork for assessing transport distances based on clast roundness which may serve as a baseline for future research to build upon. I would recommend accepting the paper after a few corrections have been made based on the following minor comments.
- AR: We appreciate your interest and the summary of our work. We agree with the points you mentioned above. Thank you once again for taking the time to review our work.

**1.1. General Comments**

- **RC:** Consider using larger fonts for headings and subheadings. Personally, it helps me find where sections begin and end. The main headings in particular should be bigger to stand out from the rest of the text.
- AR: We also prefer larger fonts for headings and subheadings. While preparing this manuscript, We followed the Overleaf template provided by Esurf, and did not modify the font settings therein. However, upon acceptance for publication, we anticipate that the paper will be formatted accordingly.
- RC: In section 4.1, it is stated (on lines 362-364) that the number of cycles of uplift, erosion and redeposition of conglomerate pebbles can be addressed using the model, but unless I am missing something, I can't

see this discussed anywhere else in the paper. Evidence for recycling of pebbles in the Karnali River is provided later in the section, stating that some pebbles in the modern river had been rounded to an equivalent of 1472 km of transport despite the length of the river only being 660 km. This would then imply that recycling was responsible for the other 800 km equivalent of abrasion. However, nowhere is this result translated into an estimate of how many cycles of uplift, erosion and re-deposition have occurred. Furthermore, it is not stated how the model could be used to provide such an estimate.

- AR: We acknowledge this comment, and we have revised this section to address the concerns raised here. We removed lines 359 to 367 from the original manuscript and incorporated additional text after line 375 in the revised manuscript.
- AR: Reviewer 1 has also made a similar comment, so this response is the same as the one we provided for Reviewer 1.
- AR: The text that has been removed from the original manuscript (line 359-367):

The recycled pebbles from the Upper Siwaliks conglomerates may have experienced multiple cycles of deposition, tectonic uplift and re-deposition as the proximal foreland basin is incorporated into the thrust wedge. The process of how thrust wedges, consisting of accredited sediments from the foreland basin, are sourced and then accredited back into the wedge, with several cycles of such, is outlined in Sinclair2011. In our study area, we do not know how many such cycles of uplift, erosion and re-deposition the conglomerate pebbles have experienced due to the ongoing shortening across the Himalaya. We can address these questions using our new rounding model. This is not without pitfalls, as the effect of weathering from the time of the deposition of conglomerates of Upper Siwaliks can affect its resistance to abrasion once it is re-entrained. However, the difference in roundness among recycled and non-recycled pebbles can still be used to consider the paleo-transport distance of pebbles that are considered to have been recycled.

AR: The text that has been added after line 374 in Sect. 4.1 Recycled Modern Pebbles (Revised manuscript):

Using gravel flux calculations and clast analyses, Quick et al. (2019) suggested that quartzite clasts deposited in the foreland basin had experienced at least one round of recycling along the Karnali River. DeCelles et al. (1998) made a similar observation in the Siwaliks sediment further west from the Karnali River, where they found evidence of two rounds of sediment recycling. The interpreted transport distance from our roundness model is greater than the length of the studied modern river, which is consistent with sediment recycling in this setting. In the Himalayan foreland basin, the rapid subsidence of the proximal basin keeps the gravel-sand transition boundary close (10-20 km) to the active front (Dingle et al., 2016). The proximity of the gravel units close to the active deformation front results in them being vulnerable to accretion back into the thrust wedge. Ongoing rock uplift and exhumation of the accreted gravels (now conglomeratic rock) results in renewed fluvial transport of clasts and deposition at the younger gravel units in front of the deformation front (Sinclair, 2011; Sinclair et al., 2018).

The depth of the décollement and the shortening rate control the thickening of the wedge (Dal Zilio et al., 2020) and, consequently, the cycles of accretion and exposure of such gravel stratigraphy. The current shortening rate at the front of the Himalaya is 17-20 mm per year (Bilham et al., 1997; Mugnier et al., 1999; Jouanne et al., 2004). The sediments of the Siwaliks (the source of the recycled pebbles in this study) were diachronously deposited from 14.6 to 1.8 Ma (Mugnier et al., 1999). Assuming constant shortening rates from the time of deposition of the Siwaliks sediment, there has

been approximately 250-300 km convergence in last 15 Ma. It is difficult to estimate with precision the extra distance that pebbles could potentially travel through a given number of cycles of recycling in such a context, as the estimate will also depend on the width of the exposure of Siwaliks gravel and on the distance at which these units are exposed upstream of new, emerging mountain fronts; the latter is found to be highly variable (e.g., Quick et al. (2019)). However, this calculation can help us bracket the extra distance that pebbles can travel through one or more cycles of recycling over the last 15 Ma, i.e., tens to a few hundreds of km. The average transport distance of the Karnali River pebbles based on our model is about 860 km, whereas the length of the modern Karnali River from its channel head is only 660 km. This provides a minimum estimate for the recycling distance of 200 km, which is not inconsistent with the calculation of potential recycling distance based on convergence rates. Although these comparisons suggest evidence of recycling, this study cannot determine the number of rounds of recycling or the amount of shortening due primarily to uncertainties in the length of the river channels that drain the Siwaliks, as many of such channels tend to run parallel to the strike of the structures.

- RC: In my opinion, the placement of figures is sometimes a bit odd. Figure 1 is mentioned one time in the text, this being at the top of page 4, but the figure itself is shown at the bottom of page 5, far away from the mention in-text. Figures 10 and 11 share a similar issue, but more egregiously so. In this case, the first in-text mentions of figures 10 and 11 are found near the bottom of page 19, but the figures appear much later. Figure 10 is shown at the top of page 21 while figure 11 is all the way down on page 22. Since figure 10 is mentioned many times above and below the figure it is not a significant problem, however, figure 11 is only mentioned on page 19 and nowhere else, making its placement several pages below (in the Discussion section rather than Results) rather confusing. The issue with having figures placed far from their in-text mentions is that it forces the reader to scroll back and forth through the document while reading, sometimes over several pages, which is detrimental to the reading experience. Of course, a perfect layout is not always possible, but in my opinion the article would be improved if the back-and-forth scrolling could be reduced, perhaps by making some figures smaller and/or moving them around.
- AR: We apologize for not giving sufficient attention to the proper sequencing of reference to figures in the manuscript. We have now made efforts to ensure that figures appear in their relevant positions immediately after being mentioned in the text.

**1.2.** Line comments:**

- RC: L13-14: In the abstract it is mentioned that the roundness coefficient for granite pebbles is eight times that of quartz pebbles, however, it is claimed on lines 342-343 and in the figure 7 description that the roundness coefficient for granite pebbles is seven times that of quartz pebbles. Whichever of these two stated numbers is more accurate should be used consistently throughout the paper.
- AR: We apologize for the inconsistencies. The comparison of the roundness coefficient between quartzite and granite has been reviewed and standardized throughout the entire manuscript. The correct value is nine times.
- AR: Line 14 in the revised manuscript.

Our field data suggest that the roundness coefficient for granite pebbles is eight nine times that of quartzite pebbles.

AR: Line 345 in the revised manuscript.

the granite's  $\lambda$  is approximately seven nine times that of quartzite.

AR: Figure 8 in the revised manuscript.

The roundness coefficient of granite is around seven nine times that of quartzite.

**RC: L115: "... will likely influence how they pebbles round" Typo, 'they' should be 'the'.**

AR: We have excluded this line, as we have reorganized the introduction section in accordance with the suggestions provided by the Reviewer #2.

**RC: L119: Informal language, usage of "don't" instead of "do not".**

AR: L119 in the original manuscript corresponds to line 70 in the revised manuscript.

A study in an Alpine river also showed that the water discharge and flow strength don't do not exert the main control on the shape and size of fluvial pebbles (Litty and Schlunegger, 2017).

**RC: L151: I'm unsure why the spacing above and below equation 1 is uneven. Equation 4 clearly has even spacing, while I can't tell for sure with equations 2 and 3.**

- AR: As mentioned earlier, we used the Overleaf template provided by ESurf. We tried with several codes (such as ignorespace, par, and noindent) but were unable to effectively control the spaces before and after equations. Nevertheless, upon acceptance for publication, we expect that the paper will be formatted appropriately.
- RC: L175: Unnecessary space before the '' in "( 100 m)".
- AR: L175 in the original manuscript corresponds to line 171 in the revised manuscript.

The Banganga River contains two thick  $(\sim 100 \text{ m}) (\sim 100 \text{ m})$  quartzite units near the headwaters of the catchment.

**RC: L220: Missing comma after "For example".**

AR: L220 in the original manuscript corresponds to line 219 in the revised manuscript.

For example example, Cassel et al. (2018) used a flat surface of  $1 \text{ m}^2$  with a red background to photograph the pebbles in the field.

**RC: L280-281 L283: Almost the exact same sentence ("Granite pebbles are rounder than quartzite ones when comparing the percentiles across lithologies") is repeated twice. Was the first instance of the sentence supposed to be removed?**

AR: L280 in the original manuscript corresponds to line 279 in the revised manuscript.

All trends show that the roundness of every percentile, including the median, increases downstream (Figure 4c and d). Granite pebbles are also rounder than quartzite ones when comparing the percentiles across lithologies.

**RC: L349: Given the context, the word 'latter' should be used instead of 'later'.**

AR: L349 in the original manuscript corresponds to line 353 in the revised manuscript.

Because the later latter category contains recycled clasts that may have gone through one or more cycles of transport, deposition, and re-entrainment, these pebbles will tend to have greater roundness than pebbles sourced from bedrock exposed in the catchment area.

- RC: L360-362: Consider rewording this sentence to make it easier to follow.
- AR: This sentence has now been removed as we have rephrased this whole paragraph about recycling.
- **RC:** L367: There is a space before the '.'.
- AR: This sentence has now been removed as we have rephrased this whole paragraph about recycling.
- RC: L368: "... in this region consists thick..." I believe "consists of" would be correct here.
- AR: L368 in the original manuscript corresponds to line 363 in the revised manuscript.

The Sub-Himalaya in this region consists consists of thick (several tens of meters) Miocene-Pliocene conglomerate beds comprising clasts of quartzite, marble, schist, phyllite, dolomite and limestone.

- **RC:** *L408: The line ends with a quotation mark, likely a typo.*
- AR: L408 in the original manuscript corresponds to line 423 in the revised manuscript.

through the present Kathmandu Basin."

- RC: L435: Given the wording of the sentence, the reference should not be in brackets but rather part of the text.
- AR: L435 in the original manuscript corresponds to line 483 in the revised manuscript.

of rest, as described by (Lajeunesse et al., 2010) Lajeunesse et al. (2010)

- RC: L445: There is an extra space before the closing bracket in "(IRn )".
- AR: L445 in the original manuscript corresponds to line 509 in the revised manuscript.

However, it is noteworthy that the parameter  $(IR_n)$   $(IR_n)$  used in this model to represent the roundness is independent of sphericity.

- RC: L456: The sentence "We present the applicability of new roundness model to the ancient and modern sediments" doesn't quite work. Did you mean to say "... of a new..."?
- AR: L456 in the original manuscript corresponds to line 512 in the revised manuscript.

We present the applicability of new roundness model to the apply our new roundness model to ancient and modern sediments.

- **RC:** I hope my comments prove helpful and I wish you the best of luck on your research.
- AR: Yes, these comments are truly helpful, and we sincerely appreciate your thoroughness in identifying typos and other issues. Thank you so much.

**2. RC1: Laure Guerit**

**AR: Dear Laure Guerit,**

We appreciate the valuable feedback and suggestions you have provided, greatly enhancing the quality of our manuscript. Base on your thorough review, we have attached a detailed response below.

Kind regards, Pokhrel et al.

**2.1. General Comments**

- RC: This is my review of the manuscript entitled "Downstream rounding rate of pebbles in the Himalaya" submitted by Pokhrel et al for publication in Earth Surface Dynamics. In this work, the authors propose a relationship between the the roundness of coarse sediments found in rivers and transport distance. The manuscript is well written and presented, and supported by adequate and readable figures. The idea is not new but the authors developed a new formalism that allow for quantification. They show how this model can be used to reconstruct transport distances of old deposits, leading to discussion about sediment recycling and drainage reorganisation. Overall, I think this is a great work with large potential for the community that should be published. However, I think that the current manuscript requires some extra work before publication. First, I feel that the manuscript needs some reorganisation (for example, some methods appear as Results, or Discussion is rather a conclusion). Second, I missed some discussions about the implications of this work. Please find below my detailed comments that will hopefully help. I have no doubt the authors can address my major concerns so I'm looking forward to reading this manuscript in Esurf.
- AR: We appreciate your concise summary of our work. We agree that the concept of a non-linear relationship between roundness and transport distance isn't new (explained in lines 50-59 in the revised manuscript), and are glad that you recognize how our new roundness model can help quantify the transport distance of coarser fluvial sediments over a long distance. About the reorganization of the manuscript, we also feel that the "Introduction" and "Discussion" sections need some reorganization, following comments by all reviewers. However, we would like to keep the Results section as it is. We agree that subsection 3.1 (Downstream Changes in Roundness and New Roundness Model) and subsection 3.2 (Derivation of Rounding Curves and Coefficients for our Granite and Quartzite Pebbles) of section 3 (Results) are actually methods. However, the new model and the approach used to derive rounding curves and coefficients are entirely built on the results of our analysis of roundness percentiles. It would be very challenging to convey to the reader the idea of a new model and shifting percentiles curves (Figure 7 in the revised manuscript) which gave us the idea of the model. Hence, even though it is a method, it is also a result that we arrived at after analyzing the field data. Therefore, we would like to keep the Results section as it is. We have now added the following text at the end of the Motivation section to justify our choice:

AR:

Details of our roundness model, which mirrors Sternberg's law of mass loss (Sternberg, 1875), are provided after presentation of the roundness data collected along the two Himalayan rivers, as these data are needed to contextualise the model.

**2.2. Introduction**

- RC: I found these pages interesting, however, they look a bit like a report rather than a paper introduction. In the current form, it is quite difficult to identify what is already known,/done how it is used and what are the current flaws that motivate this study. I strongly recommend to shorten this part in order to be more focus and somehow more conclusive about what is known and not known on the relationship between roundness and transport distance. This would really highlight the relevancy of this work.
- AR: Thank you for this advice. We have revised the introduction. Reviewer 2 also made suggestions as to how to rework the introduction, which we followed. We refer to the response to reviewer 2 for additional details on the changes that were made.

**RC:** Grain size is also used to discuss transport dynamics so it could be interested to discuss briefly what can be done with it and why shape is more interesting.**

AR: Yes, it is true that size, shape, and roundness are usually associated with each other when discussing sediment transport dynamics. We don't think shape is "more interesting", but this is a parameter that we decided to focus on based on recent publications. There is a debate about how size and roundness co-evolve and whether they control each other (Domokos et al., 2014). We briefly discussed this in lines 60-64 (in the revised manuscript). Additionally, Quick (2021) analyzed the relationship between roundness (using the same roundness parameter as used in this study), grain size (b-axis), and axis ratio (b/a) in the same setting as this study. However, Quick (2021) demonstrated no correlation between those parameters (as included in the Discussion section, lines 522-524 in the revised manuscript). Therefore, in this paper, we remain focused on the downstream rounding of pebbles and the estimation of transport distance, although we acknowledge the importance of grain size in sediment transport dynamics.

**2.3. Materials and Methods**

- RC: From section 2.1, I didn't really understand if you use a metric that has been developed by others (as suggested by the various references) or if you add something new (as suggested by line 159).
- AR: We use the metric known as Normalized Isoperimetric Ratio, as introduced by Quick et al. (2019). We believe the text below (lines 151-153 in the revised manuscript) states this clearly, but have added text to justify the use of this metric over others (see next comment).

**AR:**

Quick et al. (2019) found that the maximum IR a pebble can achieve decreases with decreasing axis ratio. They developed a 'Normalized Isoperimetric Ratio' ( $IR_n$ ) designed to remove any dependency on elongation, and only measure the angularity (or roundness) component from the IR

**RC:** In addition, I didn't understand the motivations to use this definition of roundness more than another. I think this could be clarified in the revised introduction.**

AR: The motivation behind using Normalized Isoperimetric Ratio is based on the studies conducted by Roussillon et al. (2009) and Quick et al. (2019). In their research, Roussillon et al. (2009) compared various geometric parameters that characterize roundness and identified circularity as a more powerful metric for measuring roundness than other metrics. Both circularity and Isoperimetric ratio utilize the area and perimeter of the shape to characterize roundness, but they also include elongation in their measurements (Roussillon et al. 2009; Quick et al. 2019). Consequently, to exclusively quantify roundness in pebble shapes, we employ the Normalized Isoperimetric Ratio, as first proposed by Quick et al. (2019). In Section 2.1 (Choice of Shape

Index), we now add text (lines 133-146 in the revised manuscript) expanding on our motivations for choosing this definition of roundness over others (including a new figure to explain better why the Isoperimetric Ratio needs to be normalized).

- RC: Similarly, if I understood correctly, you didn't develop new algorithm for image processing but you used options that are already available in ImageJ. It is totally fine but it is not that clear from the current section 2.4.
- AR: Yes, it is true that we didn't develop a new algorithm for image processing in ImageJ. However, we propose a workflow, starting from how to photograph the pebbles, followed by image processing in ImageJ and calculations in a GIS environment. We have revised the text in section 2.4 for clarity.
- AR: Line 235 in the revised manuscript has now been rephrased.

The basic principle of our automatic method followed in this study is to read the pebble silhouette automatically by a software using a colour threshold.

- RC: Also, there are several methods available to segment grains and extract geometrical information (see for example BaseGrain, PebblesCount, G3point) so this could be explored a bit in the introduction.
- AR: We initially attempted to use PebblesCount; however, we discovered that ImageJ is more convenient for us to work with. We have already incorporated PebbleCounts into our manuscript and have now included the other methods in the Introduction section (line 107 in the revised manuscript).

**RC: Some might avoid the use of 2 softwares including one commercial (ImageJ and ArcGIS) and make your methodology more open access.**

- AR: We applied ImageJ, a non-commercial open-access software, for image processing. However, for the conversion of raster images into vector polygons, any open-access GIS environment can be used. In this project, we opted ArcGIS, but the work could equally be done using other open access GIS software.
- AR: The text has been added in line 264 (revised manuscript).

This step provides all the measurements necessary to calculate the parameter used as a measure of roundness in this study. Although we used ArcGIS for the conversion of raster images into vector polygons, the work could equally be done using other open-access GIS software/packages.

**RC: In addition, I feel that the level of details is sometimes too high (for example, I don't think you need to mention that you used HCl to identify rocks - again, this gives the feeling of a report rather than a paper).**

- AR: We intentionally mentioned the use of HCl, which is important for identifying the rocks in the study catchments where there is a transitional contact between the siliceous rock (Quartzite) and carbonate rock (Limestone and Dolomite). However, we removed this line to balance the level of details provided here.
- AR: The text has been removed from lines 203-204 in the original manuscript.

This area lacks granite in the source region, so only the pebbles of quartzite rock are collected from this catchment. We used 10% dilute Hydro-chloride acid (HCl) to differentiate the pebbles sourced from greyish siliceous carbonate rocks (limestone and dolomite) from those derived from quartzite. We

carefully examined pebbles based on texture and mineralogy using a hand lens, thus we are confident all our sampled pebbles in the Banganga River are indeed quartzite and not some other rock type.

**RC: There are also some repetitions (for example, l. 208 is a repetition of L. 184, l. 211 is a repetition of l. 193).**

AR: Lines 206-207 (in the revised manuscript) have been modified to eliminate repetition. Likewise, line 211 in the revised manuscript has also been revised.

We applied a similar sampling procedure in the Rapti catchment, south of the Kathmandu. In the Rapti, we sampled granite pebbles rather than quartzite, as explained in Sect. 2.2. The field identification of granite pebbles is easier than identification of quartzite pebbles as there are no other rocks with igneous texture exposed in this study catchment.

Upon arriving at a potential sampling site, we first assessed whether the gravel bar was close to the active channel (i.e., not a terrace) and not influenced by human activities .

**2.4. Results**

**RC: As a general comment, a lot of this section sounds like methods and should be presented as such, or at least in a part dedicated to "Modelling" for example.**

- AR: We address this comment by reiterating some parts of the author's response to general comments. We agree that subsection 3.1 (Downstream Changes in Roundness and New Roundness Model) and subsection 3.2 (Derivation of Rounding Curves and Coefficients for our Granite and Quartzite Pebbles) of section 3 (Results) in the manuscript are actually methods. However, it is also true that if we hadn't observed the downstream trend of varied slope for different percentile roundness (Figure 5 (c) and (d)), we might not have come up with the idea of transferring a higher percentile to the downstream to generate the roundness curve for a greater distance (Figure 7). It would be very difficult to convey to the reader how we developed the new model, as well as the method we developed to derive the theoretical curves, without showing the data first. For this reason, we believe it is important to keep these sections where they are, and have added text to justify this accordingly (see earlier response to general comment).
- **RC:** I like your interpretation that the percentiles of the roundness distributions are families of pebbles evolving along the river. However, I'm a bit concerned by the limited number of points in the lower quantiles. With on average 30 samples for the granite, the 5th percentile might be represented by 1 grain, a few at best. I would recommend to have a minimum number of samples (maybe 5?) in a percentile before to use it.
- AR: Yes, we agree and accept that this limitation is inherent in our work. Unfortunately, we feel that such an approach would lead to discarding a lot of the data the  $95^{th}$  percentiles would also be concerned, as their value will also be driven by the top 5% of the sampled grains, and each interval will also be influenced by a few grains. The choice of a minimum number of grains would also be arbitrary. Instead, we would like to keep all data as it is and add a statement in the discussion regarding this limitation. We don't have more data points due to the limited availability of granite clasts along the channel bars (as shown in Figure 3 (d) granite is not the predominant rock type exposed in the catchment area), but we feel the changes in the distributions are real (Figure 5 (a)). Hopefully the work generates enough interest that others can apply the same strategy in other places to test the model.
- AR: The text from line 445 to 454 has been added to the revised manuscript.

One of the main uncertainties comes from the fact the number of grains measured at each site varies (Figure 5 (a) and (b)). In the case of granite pebbles, we measured an average of 30 individual pebbles at each site. In such cases, the lower and higher percentiles may be represented by a single to a few grains. The maximum number of grains at each site was controlled by availability (as shown in Figure 3 (d); granite is not the predominant rock type exposed in the catchment area), and we accept that this limitation is inherent to our work. We could have discarded percentile values that are driven by a number of clasts smaller by a threshold. However, the choice of a threshold value would have been arbitrary and such an approach would lead to discarding much of the data in the higher ( $95^{th}$ ) and lower ( $5^{th}$ ) percentiles; in addition, all intervals will also be influenced by a few grains. We therefore opt to present and use all the data available. We believe that the changes in the downstream distributions of the roundness data are real (Figure 5 (a)) and hope that this work encourages other researchers to use the same strategy in other locations with better constraints to test the model.

- RC: I was a bit confused at first by the statement line 287 as panel c on Figure 4 shows that angular grains round faster than the others. I assume you mean that whatever the initial angularity, once two grains have the same roundness, they continue to round at the same rate. I suggest to rephrase a bit this part to make it more clear.
- AR: We understand that this was confusing. What you wrote is correct. Lines 285-288 (in the revised manuscript) have been revised for clarity.

If we make the assumption that it is impossible for a given pebble to round downstream faster than another (i.e., a pebble starting with a lower  $IR_n$  than another will always have a lower  $IR_n$  than the other if they travel the same distance), then we can assume that each percentile represents a population that evolves downstream, and the linear fits represent sections of an asymptotic trend that occurs over much longer distances, with a gradient decreasing rapidly as  $IR_n$  approaches the asymptote (Figure ??). if two grains have the same roundness, they will round at the same rate, then we can infer that each percentile represents a population evolving downstream. The linear fits in the graph could therefore represent sections of an asymptotic trend occurring over much longer distances, with a gradient that decreases as  $IR_n$  approaches the asymptote (see Figure 6 in the revised manuscript).

- RC: A major issue of this part is that I didn't find a definition for the transferred distance. I assume that it is the total transport distance of the grains, which is different from the distance to the headwaters to due drainage reorganisation and recycling. Please add a clear sentence about it (and I think it should be defined in the Methods, not in the Results).
- AR: The definition of 'Transferred transport distance' is now included after line 327 (revised manuscript).
- AR:

We consider the R – squared values (with vertical residue) as the evaluation metric. The term 'transferred transport distance' is a downstream distance along which the percentiles higher than the 5th percentile are shifted to a greater distance, assuming that the higher percentile represents the pebbles transported to the greater distance. Hence, this distance does not represent the distance from the channel head but instead represents the required transported distance for pebbles to achieve greater roundness, beginning from an initial roundness at the distance d = 0. In this model, this initial

roundness is set by the lowest percentile data ( $5^{th}$  percentile) for which distance has not been shifted and the prefactor k

- RC: On Figure 7, you show the theoretical rounding curves for the two populations of grains. I really like it but I have a few questions:
- RC: do you think it is possible to derive an enveloppe, rather than a curve, in order to estimate uncertainties ?
- AR: An uncertainty envelope, representing a 95% confidence interval for both curves, has now been added (Figure 8 in the revised manuscript).

Figure 1: (Figure 8 in revised manuscript) Theoretical roundness curve for granite (green) and quartzite (blue) derived from the optimisation method and regression of  $ln(1 - IR_n) = f(d)$  field data. Each marker and colour represents field roundness data. The roundness coefficient of granite is about nine times that of quartzite.

AR: The following line has been added in the figure caption.

An uncertainty envelope of a 95% confidence interval for both curves is calculated using the standard error of the sample mean.

- RC: -The x-axis goes up to 2000 km, yet, it is quite unlikely that pebbles survive that long (see for example Dingle et al, 2017). Therefore, what was the motivation for such a long x-axis and in which configuration do you expect pebbles to live that far?
- AR: While this is true for most pebbles, quartzite pebbles are extremely resistant and can survive transport distances of thousands of km (as mentioned by Dingle et al. 2017). A pebble abrading at 0.1 % mass loss per kilometre (which is a realistic value for quartzite, see Attal and Lave 2009 for example) will lose 86% of its mass in 2000 km. Many rivers in the world have lengths of a few thousand kilometres, so we think it makes sense to present the complete curve in this context. In the Nepal Himalaya, the length of the longest river from the channel head to the gravel-sand transition is approximately 1000 km. And sediment recycling can allow pebbles to cover longer distances. For example, the sediment transport distance for recycled pebbles along the Karnali River (location marked with a red star in Figure 8(a)), calculated using our model, is approximately 1500 km, based on pebble roundness data from Quick et al. (2019) and Quick (2021). Although we calculated the maximum transport distance using the 95th percentile value, some individual pebbles have roundness values up to 0.99. According to our model, a distance of approximately 2000 km is where quartzite pebbles achieve perfect roundness, which is why we included such a long x-axis.
- AR: The following paragraph has now been added after line 487 (revised manuscript) in the discussion section to address the issue raised above.
- AR:

In the Nepal Himalaya, the length of the longest river from the channel head to the gravel-sand transition is approximately 1000 km, and many other rivers in the world have a length exceeding 1000 km. In general, most pebbles will not survive a transport distance of thousands of km; however, quartzite pebbles are extremely resistant and can survive transport distances of thousands of km (Dingle et al., 2017). Attal and Lavé (2009) document abrasion rates as low as 0.1% mass loss per km for quartzite pebbles. At this rate, a quartzite pebble will lose 86% of its mass in 2000 km. Sediment recycling can allow pebbles to cover additional distances. For example, the maximum sediment transport distance for recycled pebbles along the Karnali River (location marked with a star in Figure 3 (b)), calculated using our model, is approximately 1500 km. Although we calculated the maximum transport distance using the  $95^{th}$  percentile value, there are some other pebbles in the population which are perfectly rounded (roundness value ~1.0). According to our model, a distance of approximately 2000 km is where quartzite pebbles achieve perfect roundness (Figure 8).

**2.5. Section 4**

- RC: I really like this part, but I would have appreciate to see it as Results followed by a proper discussion. I understand the potential impact of recycling on total transport distance, however, the way it is shown in Figure 8 will not increase the distance. In order for recycling to have a significant impact of total distance with respect to the position where the grain is found, you need to have shortening. This leads to two comments:
- AR: We have changed the figure (Figure 9 in the revised manuscript) to better convey the importance of shortening to increase transport distance.

**RC:** - The results should be discussed with respect to the known shortening rates in the area. - Do you think your approach could be used to estimate shortening from the difference between position and total transport distance?**

- AR: This is not straightforward, as it requires knowledge of the length of the river systems, width of conglomerate exposures, and distance at which the rocks are exposed from the mountain front. As shown in Quick et al. (2019), these parameters do change a lot along the length of the Himalaya. A new paragraph has now been included at the end of section 4.1, and lines 359 to 367 (original manuscript) have been removed. This modification takes into account the known shortening rate in the area and the transport distance.
- AR: PC #1 has also made a similar comment, so please refer to the section 'response to PC #1' or section 4.1 (lines 375-399) in the revised manuscript.
- RC: It is interesting to compare with the geological map, but I think you could better articulate the two approaches. Somehow, the map confirms your findings but it would have been easier to simply look at the map and identify that the grains must come from an area that is no longer part of the catchment. I think you could explain a it more the advantages of your approach in this kind of context (this comment is related to my first comment on the Introduction).
- AR: It is true that the geological map confirms our finding. However, the geological map alone cannot provide information about the distance or the source area of the sediments, particularly in regions where identical lithologies exist in different stratigraphic positions; interpretation of the sources of sediment for the proto-Bagmati has been debated for a long time. For example, in Figure 11 (revised manuscript), the geological map indicates two potential sediment source areas: Paleozoic granite south of the Kathmandu Basin and Tertiary granite north of the basin. By looking at the geological map alone, we can't define the source area for the granite clasts that we observed at the base of the Kathmandu Basin. In addition, mapped units may have been exposed in very different places 2 Ma ago, as high erosion rates such as those found in the Himalaya can lead to the removal of km of rocks over the course of millions of years. Therefore, we believe that our roundness model helps narrow down the sediment source area for the paleo-river channels, offering a more advantageous approach than relying solely on the geological map.
- AR: A paragraph has been added at the end of the subsection (lines 424-433) "Pebbles from the stratigraphic record" to provide further explanation.
- AR:

While the location of mapped units may have been sufficient to conclude that the granite clasts were sourced from outside the basin, modern geological maps are not always entirely reliable to identify the source of clasts, particularly in regions where identical lithologies exist in different stratigraphic positions. For example, in Figure 11, the geological map indicates two potential sediment source areas: Paleozoic granite south of the Kathmandu Basin and Tertiary granite north of the basin in the High Himalaya. By relying solely on the geological map, we cannot definitively determine the source area for the granite clasts observed at the base of the Kathmandu Basin. In addition, mapped units may have been exposed in different places 2 Ma ago, as high erosion rates, such as those found in the Himalaya, can result in the removal of kilometres of rocks over millions of years. Here, the high transport distance calculated for granite and quartzite pebbles in the Kathmandu Basin suggests that they were deposited around 2.5 Ma ago by trans-Himalayan rivers, demonstrating the benefits of our roundness model to narrow down the sediment source area for paleo-river channels.

**2.6. Discussion**

RC: This section is more a long conclusion. It's fine but in consequence, I missed a proper discussion on the results:

**RC:** is there a relationship between the initial size and the initial angularity?**

AR: In the discussion section, specifically between lines 473-487, we explain the mechanisms for the production and deposition of sediments with varying degrees of roundness along the river. It is probable that this explanation also sheds light on the coexistence of sediments with different grain sizes within the same location. However, our exploration did not extensively cover the relationship between size and angularity. The motivation behind this is rooted in the findings of Quick (2021), who demonstrated no correlation between downstream fining and rounding of grains, as mentioned in lines 522-527.

**RC: How can rock breaking affect the rounding curve?**

AR: We explain this from lines 504 to 511 in the revised manuscript. The roundness metric employed in this model is independent of sphericity, indicating that pebbles do not need to be perfectly spherical to attain maximum roundness. In this context, rock breaking may enhance angularity (a round pebble will see its roundness decrease dramatically if it is broken in two). If the rock-breaking process becomes dominant downstream, our downstream rounding model may no longer be valid.

**RC:** More specific to the results on the Himalaya: are they evidences for major drainage reorganization as suggested by the transport distance?**

AR: Yes, there are some speculations (Hagen, 1969) regarding major drainage reorganization in central Nepal, as mentioned in line 423 (revised manuscript).

**RC:** What about the transferred distances?**

AR: The definition has now been clarified (See earlier response).

**RC:** How can we use this approach elsewhere? It would really strengthen the manuscript.**

- AR: This is indeed a great question, and we are keen to broaden its applicability. The prefactor and roundness coefficients presented in this paper are derived from two small catchments of Himalayan rivers. We believe that future research comparing the prefactor and roundness coefficients from catchments in different geomorphic settings will contribute to the comprehensive development of this approach, and we hope that this work encourages other researchers to carry out similar studies in other settings.
- AR: We have added a paragraph at the end of the discussion section, which will help other researchers to follow the approach. We refer to the discussion section in the revised manuscript.

**2.7. Conclusions**

- **RC:** Please make a it more clear what is new results from this study and what is not.
- AR: We have rephrased the entire conclusion section; therefore, we refer to the conclusion section in the revised manuscript.

**2.8. Minor Comments**

RC: •1. 14 "eight times that of quartzite" : it is written "seven" in the caption of Figure 7. Please correct.

AR: The comparison of the roundness coefficient between quartzite and granite has been reviewed and standardized throughout the entire manuscript. There is now a nine-fold difference between the two coefficients.

Our field data suggest that the roundness coefficient for granite pebbles is around seven nine times that of quartzite pebbles.

**RC: •1. 66: Feher et al is a preprint that has been withdrawn by the authors. It can't be used as a reference.**

AR: The citation of Feher et al. (2020) has now been removed from the text.

which can measure traditional, mathematically complex and common geometric shape parameters. Fehér et al. (2020) demonstrated the effectiveness of 3D laser scanning of beach pebbles by comparing the results with a hand-measured set of pebbles. Thus, advances in technology are making automatic extraction of shape parameters possible.

**RC: • l. 191 missing (before "Mudd"**

AR: Line 187 in the revised manuscript.

We extracted flow distance from the channel head using the LSDTopoTools software Mudd et al. (2022). (Mudd et al., 2022).

**RC: • Lines 237 to 266 should be in Appendix rather than in the main text.**

AR: Initially, we considered including this text in the appendix. However, upon further reflection, we realized that it would be beneficial to incorporate it into the main text. Through our exploration of the literature for this manuscript, we concluded that this information would be valuable for those intending to utilize the method we propose, as we assert in this manuscript that we present a comprehensive workflow for the measurement and analysis of pebble roundness. Furthermore, the length of the manuscript is not excessively long. Therefore, we believe it is important to retain this information in the main text.

**RC: • Fig. 4 The text at the top of each panel is quite close to the boxes. If possible, consider adding a bit of space. Missing captions about the colors on panels a and b (it does not seem to carry any information but maybe it can be use also on Figure 2?)**

AR: The colors on panels a and b of Figure 4 (Figure 5 in the revised manuscript) don't carry any information. Now, figure has been updated by considering the same colour for the box plots in panels a and b. We refer to the revised manuscript.

**RC: • l. 371 extra capital s "Sample"**

AR: 1. 371 in the original manuscript corresponds to line 366 in the revised manuscript.

The Sample site is located in the Indo-Gangetic plain which consist of a full mixture of sediments from Higher to Lesser Himalaya and Sub-Himalaya (Sample (sample site in Figure 8).

**RC: • Fig. 9 missing "a" in quartzite**

AR: Figure 9 (Figure 10 in the revised manuscript) has been updated with the correct spelling of "quartzite".

**RC: • Fig. 10 what are A and B for? Missing space in the caption between Basin and Note.**

AR: In Figure 10 (Figure 11 in the revised manuscript), A is the channel head, and B is the location of the channel to which the distance of the modern Bagmati River is measured in order to compare the length with the transport distance estimated from the pebble roundness. The description of A and B has now been included in the Figure caption.

both granites (TGr and PGr) are outside of the catchment area of the modern Bagmati River in Kathmandu. In the map, A represents the channel head, and B indicates the location of the channel where the distance to the modern Bagmati River is measured.

AR: The missing space in the caption between "Basin" and "note" has now been corrected.

granite pebbles found at the base of the Kathmandu Basin.Note: Basin. Note: both granites

**RC:** • *l.* 409 extra "**

AR: 1. 409 in the original manuscript corresponds to line 423 in the revised manuscript.

the Bagmati River supports the previously hypothesised extensive drainage network (Hagen, 1969) through the present Kathmandu Basin."

**3. RC2**

**AR: Dear Anonymous Reviewer,**

Thank you very much for your thorough review and your helpful comments and suggestions, which have significantly contributed to improving our manuscript. As your review is very comprehensive, we have provided a detailed response below.

Kind regards, Pokhrel et al.

**3.1. General Comments**

- RC: The authors present the paper "Downstream rounding rate of pebbles in the Himalaya", which outlines a method for measuring roundness of pebbles, as well as proposes a model for relating pebble roundness to transport distance. They specifically address challenges with previous methods of measuring pebble shape parameters, and identify a method for automatic and repeatable extraction of these shape parameters (namely isoperimetric ratio) from 2D photographs of pebbles using publicly available software. This shape characterization method was applied to pebbles collected from two rivers in Nepal, as well as clasts found within conglomerate deposits in a similar region, which were then used to calculate rounding curves for two rock types within these watersheds. By determining transport distance of historical clasts, the authors drew conclusions about the length of paleo rivers in the Himalayas.
- AR: We appreciate your concise summary of our work, and we agree with your points. Thank you for taking the time to review our project.
- RC: This work contributes a useful approach for relating clast shape to transport distance, especially for river systems where data collection is limited to shorter distances or a smaller number of field sites, or for applications in paleoenvironment reconstruction. Additionally, from my exposure to the pebble rounding literature, it seems that others emphasize rounding as a function of mass loss in order to draw general conclusions, while this study seems more practical for direct use in specific watersheds. However, I do think that this work could be even more impactful if the authors connected their results to the larger context of universal pebble rounding behavior, such that their work could readily be applied to other geomorphic settings. Given discussion of the results within this context, as well as editing for clarity and conciseness, I think that this paper suitable for publication in ESURF and believe that it contributes to the pebble rounding literature. I hope the authors find my comments to be helpful and constructive, and I wish them the best in their research endeavors.
- AR: In this comment, we noted the suggestion to enhance the impact of this work by expanding the scope of the pebble rounding behavior within a broader geomorphic context. We thoroughly discussed how widely our roundness model and the specific values for the pre-factor and rounding coefficient can be applied in the discussion section (Section 5). We believe that the non-linear relationship between pebble roundness and transport distance is likely applicable to other fluvial geomorphic settings. However, it is important to acknowledge that the calibrated values of the prefactor and roundness coefficient may vary for different geomorphic settings. For example, the granite pebbles measured in this project are enriched with feldspar and mica grains, making them susceptible to rapid weathering and abrasion. Conversely, granite pebbles from another region may be less rich in feldspar and mica grains, rendering them more resistant to weathering and abrasion. Consequently, these variations in mineral composition could necessitate different pre-factor and roundness coefficient values compared to those proposed in this paper.

- AR: lines after 499 (in the revised manuscript) have now been rephrased
- AR:

The quartzite pebbles are sourced from a massive bed of mono-mineralic (quartz) quartzite while with no or slight degree of weathering. The granite pebbles are rich in feldspar and mica minerals, making them susceptible to rapid weathering and abrasion. Conversely, granite pebbles from another region may be less rich in feldspar and mica, rendering them more resistant to weathering and abrasion. Consequently, variations in mineral composition may imply different pre-factor and roundness coefficient values compared to those proposed in this paper. Therefore, the values obtained in this study should not be used as universal values for all granite and quartzite pebbles.

**3.2. Specific Comments**

- **RC:** Broadly, I think that the organization of the introduction (Section 1) can be improved and streamlined. Section 1.1 includes discussion on both previous pebble abrasion research, as well as shape parameters that have traditionally been used to quantify pebble roundness. Subsequently, Section 1.2 discusses shape parameters in depth, then Section 1.3 discusses controls on pebble shape and summarizes previous research in more depth. I would recommend combining the first two paragraphs of Section 1.1 with Section 1.3 and presenting this information first as background on pebble abrasion processes. I would then combine the last paragraph of Section 1.1 with Section 1.2 and present this information as background on shape indices.
- AR: We also believe that this reorganization will help streamline the pebble abrasion process and the shape indices described in the introduction section. Now we have reorganized the introduction section mostly in a similar way as mentioned above. We refer to the revised manuscript.
- RC: Section 1.1, paragraphs 1 and 2 address previous research on pebble chipping/abrasion/attrition. It seems worthwhile to include a sentence or two at the outset on the definitions used in this paper, since previous studies use different terms to refer to specific breakdown mechanisms. For example, in paragraph 2, the authors write that "processes like sandblasting, chipping, and granular removal by crushing or grinding" fall under abrasion and increase pebble roundness, but then in the same sentence, use the phrase "chipping of large fragments" as a process that reduce pebble roundness. As a reader, I am slightly confused as to how chipping is defined such that is can both increase and reduce roundness, especially since chipping appears to primarily be used in the literature to describe the process of pebble rounding due to bedload transport (e.g., Novak-Szabo, et al., 2018). Additionally, I am aware that crushing or grinding tends to fall under the purview of "comminution", which primarily breaks rocks down into smaller pieces and may increase sphericity, but would not necessarily increase roundness. From my experience with the pebble abrasion literature, the terms "abrasion" and "chipping" tend to be used to describe the small-scale breaking off of edges and corners that progressively round rocks (e.g., Miller, et al., 2014; Szabo, et al., 2015; NovakSzabo, et al., 2018), while "attrition" is a more general term that could describe small or large scale breakdown (e.g., Miller and Jerolmack, 2021), and "fragmentation" is used for significant breakdown into large pieces (e.g., Novak-Szabo, et al., 2018).
- AR: For clarity and to remove confusion regarding the terminology used in this manuscript, the text in lines 36 to 42 (in the revised manuscript) has been revised as follows:

AR:

Abrasion processes like sand blasting, chipping and granular removal by crushing or grinding will increase roundness, whereas chipping of large fragments, cracking and subsequent fracturing will decrease it (Brewer and Lewin, 1993). Since the terminology used in the published literature may vary, we clarify that we use the term 'abrasion process' to broadly describe processes that lead to mass loss of grains due to energetic impact during fluvial transport (similar to what Miller and Jerolmack (2021) describe as 'attrition'). These processes include the small-scale breaking off of edges (chipping), corners and other fragments due to impacts during fluvial transport (e.g., Miller et al., 2014; Szabo et al., 2015; Novak-Szabo et al., 2018). We use the term 'fragmentation' to exclusively describe significant breakdown of a grain into large pieces (e.g., Miller and Jerolmack, 2021).

- RC: In general, I found Section 1.1 (last paragraph) and Section 1.2 to be rather lengthy in the description of shape parameters used in the literature and methods for calculating those shape parameters. For example, I think that the discussion of automated image processing methods for grain shape detection can be reduced to only methods utilized or built upon by this study. It may also help readers if you state that circularity and isoperimetric ratio are equivalent shape indices earlier in the paper, since some readers may be more familiar with the term circularity.
- AR: We appreciate your comment and have revised the text to shorten the paragraphs, balancing the level of included details. We have reorganized the introduction section mostly as suggested by both reviewers. We refer to the revised manuscript.
- RC: Section 1.3 addresses the effect of lithology on pebble rounding. I think this section could be strengthened with the addition of background on the relationship between material strength and attrition. Since rock strength is known to control rate of attrition (Sklar Dietrich, 2001; Wang, et al., 2011 Abrasion of Yardangs) and has implications for attrition mechanism (specifically for abrasion/chipping in Miller Jerolmack, 2021), I feel that this may be useful in interpreting rounding of your granite vs quartzite pebbles.
- AR: We acknowledge that the relationship between the erosion rate and tensile strength for the various rock types proposed by Sklar and Dietrich (2001) can be inferred to describe the resistant to abrasion of the pebbles. We have revised the text in the introduction (line 64-revised manuscript) section and two paragraphs (lines 455-472) in the discussion section to incorporate the above points.

**AR:**

The downstream evolution of a pebble's shape and roundness has been showed to be controlled by the initial grain size, hardness and existence of fabrics within (Kuenen, 1956; Lindsey et al., 2005), with some of these factors directly related to the lithology of the pebble itself (Kuenen, 1956; Sneed and Folk, 1958). ; Sklar and Dietrich, 2001).

**AR:**

Variations in the coefficient of roundness ( $\lambda$ ) are observed among different rock types. The  $\lambda$  for quartzite and granite pebbles from the Himalayan river were calculated using field data, and it was found that granite is significantly more susceptible to rounding than quartzite. This result is consistent with previous studies on the control of lithology on abrasion and roundness (Kuenen, 1956; Sneed and Folk, 1958; Kodama, 1994; Lindsey et al., 2005). Variations in the coefficient of roundness ( $\lambda$ ) are observed among different rock types. The  $\lambda$  for quartzite and granite pebbles from the Himalayan

river were calculated using field data, and it was found that granite is significantly more susceptible to rounding than quartzite. This result is consistent with previous studies on the control of lithology on abrasion and roundness (Kuenen, 1956; Sneed and Folk, 1958; Kodama, 1994; Lindsey et al., 2005; Sklar and Dietrich, 2001). Sklar and Dietrich (2001) used an experimental setup to study bedrock erosion rates and processes. They found a relationship between bedrock abrasion rate and the tensile strength of the bedrock slabs used in the experiments. One may wonder if similar relationships may be found between tensile strength and the rounding of pebbles (which is driven by abrasion). Based on the rock descriptions provided, the granite and quartzite pebbles used in this study are equivalent to the weathered granite (16) and quartzite (28) of Sklar and Dietrich (2001). According to the relationship illustrated by Sklar and Dietrich (2001), the tensile strength of quartzite is almost six times greater than that of weathered granite, which mirrors the greater roundness coefficient of granite, almost nine times greater than that of quartzite, observed in our work. Similarly, Sklar and Dietrich (2001) used two types of granite: a weathered granite (16) and a granite (27). They found that the tensile strength of granite is several times greater than that of weathered granite. This indicates that tensile strength varies even within the same rock types, and that the rounding coefficients will likely reflect these differences. The two parameters of our roundness model (prefactor and roundness coefficient) are therefore likely to be dependent on bedrock lithology, but also tectonics and climate that influence weathering type and rate, and dominant hillslope sediment processes. These factors will determine the distribution of initial grain size and abrasion processes, as suggested by Sklar and Dietrich (2001)

- RC: I think that the inclusion of a figure (or addition to a pre-existing figure) could improve the discussion in Section 2.1 regarding selection of the normalized isoperimetric ratio as the roundness parameter. The reader might be able to better conceptualize the isoperimetric ratio and normalized isoperimetric ratio for different pebble shapes if there was a figure showing shapes and their IR values. Figure 12 already does this for normalized isoperimetric ratio.
- AR: We appreciate the feedback, and in response, we have included a new figure (Figure 2 in the revised manuscript) to distinguish the concept between the isoperimetric ratio and the normalized isoperimetric ratio for various pebble shapes.
- RC: In Figure 2 panel b, it may help to draw the reader's eye to the location of the study catchments by outlining them in red or bolding the catchment names. Currently, the brown outline is a similar color to mid-elevations on the map.
- AR: We addressed the issue by highlighting the catchment names in bold and changing the outline colour of the catchments to red (Figure 3 in the revised manuscript).
- RC: The last three paragraphs of Section 2.4 include specific instructions for pebble shape extraction and measurement in ImageJ and ArcGIS. I feel that the explicit step-by-step instructions are unnecessary and could largely be eliminated, or included as an appendix, to shorten the paper.
- AR: Initially, we considered including this text in the appendix. However, upon further reflection, we realized that it would be beneficial to incorporate it into the main text. Through our exploration of the literature for this manuscript, we concluded that this information would be valuable for those intending to utilize the method we propose, as we assert in this manuscript that we present a comprehensive workflow for the measurement and analysis of pebble roundness. Furthermore, the length of the manuscript is not excessively long. Therefore, we believe it is important to retain this information in the main text.
- RC: Sternberg's law is first brought up in Section 3.1 (other than the abstract). Given that this is the basis

Figure 2: (Figure 2 in the revised manuscript) Illustration of the effect of elongation (b/a axis ratio) on the roundness measurement (Isoperimetric Ratio). In the figure, A is an angular (IR = 0.84) and spherical (b/a = 1.0) pebble, B is a perfectly rounded (IR = 1.0) and spherical pebble (b/a = 1.0), and C is a perfectly rounded (but  $IR \neq 1.0$ ) and elliptical (b/a = 0.5) pebble. Although the elliptical pebble (C) is perfectly rounded, its roundness is equivalent to that of the angular and spherical pebble (A) due to elongation. The solid black line in the figure represents the theoretical maximum isoperimetric ratio as a function of the axis ratio. With the use of the Normalized Isoperimetric Ratio (the roundness metric used in this study), pebbles B and C will have the same roundness, thus removing the effect of elongation and measuring only the roundness. (Figure adapted from Quick et al., 2019)

**of the proposed abrasion model, it may be appropriate to introduce Sternberg's law in the introduction section of the paper.**

AR: We agree and now, we have included Sternberg's law in the introduction section (subsection 1.3 Motivation).

AR:

Based on measurements in two Himalayan catchments with varied rock types and provenance settings, we propose a new model to relate the roundness (IRn) with the transport distance (d), that is, the distance travelled by the pebbles from their entrance point in the river system to the location where they were measured. Details of our roundness model, which mirrors Sternberg's law of mass loss (Sternberg, 1875), are provided after presentation of the roundness data collected along the two Himalayan rivers, as these data are needed to contextualise the model. We further explore the applicability of our roundness-distance relationships to estimate the distance travelled by Miocene and Pliocene sediments in the Himalaya.

RC: In Figure 4, you may want to increase font or bold the panel labels (a., b., c., d.) since they are a similar font size to the letters used for site locations.

- AR: The font size of the panel labels in Figure 4 (Figure 5 in the revised manuscript) now has been increased and provided with the bold text for better visualization.
- RC: The Figure 8 identifies a sample site along the Karnali River. When the figure is first referenced in Section 4.1, it is not immediately apparent to me whether additional field data was collected during this study, or of the same site is from Quick, et al. (2019). Later in Section 4.1, you state that the recycled pebble field data is from Quick, et al. (2019), but it may help to clarify this in the figure caption. Further, in Section 4.2, you mention more field data from the Bagmati River collected during this study. It may be relevant to briefly introduce this field site in the methods section, as well as place this site and the Quick, et al. (2019) site on the map in Figure 2.
- AR: In the caption of Figure 8 (Figure 9 in revised manuscript), the data source for the pebble roundness data of the Karnali River has been incorporated, referencing Quick (2021). A concise description of the Karnali site and the Kathmandu site has now been included in the 'Materials and Methods' section, specifically in subsection 2.2 'Study Catchment and Site Selection' following line 193 (revised manuscript). Additionally, the locations of both the Karnali Site and Kathmandu Site have been incorporated into Figure 2 (b) (Figure 3 (b) in the revised manuscript).

AR:

In addition to these two catchments, we use data from two other locations (marked by stars in Figure 3 (b)), along the Karnali River in western Nepal and the Kathmandu Basin in central Nepal. Based on these data, we discuss the applicability of our new roundness model for both modern and ancient river systems in Sect. 4. The data for the stratigraphic record of the Kathmandu Basin is collected during this study, and data on sediment recycling along the Karnali River was previously collected by Quick (2021).

- RC: In the discussion section, the authors address the coefficient of roundness, , and prefactor, k, which corresponds to initial shape of the pebbles. Since varies for the two rock types in this study, it would be interesting to address the relationship between material strength and the coefficient of roundness. There is also discussion of how the prefactor, k, varies depending on a variety of factors. Domokos, et al. (2015 Universality of fragment shapes) show that fragmented rocks have a general mass and shape distribution. Assuming that rock fragments entering upland river systems in the Himalayas were generated by energetic processes, it might be interesting to address how the prefactor could be generalized across different watersheds. Further, assuming a general size distribution for initial fragments and that bedload dominates subsequent transport, a universal mass loss curve for particle rounding by bedload can be reached (Novak-Szabo, 2018). While this universal rounding curve relies on knowing the mass of an initial particle, perhaps you can discuss how universal behaviors, along with material-specific properties, can allow you to generalize your model and determine transport distance across a variety of watersheds/conditions.
- AR: We have now included discussion about material strength and the roundness coefficient in the discussion section, building on the work by Sklar and Dietrich (2001). We have also discussed the general applicability of our findings from a broad perspective. We appreciate the suggestions provided here, which are interesting to explore, but we feel that developing the discussion in this direction would be overly speculative. We do not think we have enough information to make strong inferences regarding the universality of the processes we try to account for with our model. We have very little constraints on the grain size distribution of the initial particles released from the hillslopes and, more importantly, the fact that fragments of all size and shapes appear so close to the headwaters (see discussion about granite fragments) suggest that the starting point is

far from being uniform for all clasts, potentially because bedload transport may not be the dominant shaping process in the first km of a pebble's journey through the landscape (effect of weathering on the initial shape of granite clasts? Effect of highly energetic collisions is the clasts have started in a debris-flow?) We hint to these at the end of the discussion and hope these ideas could represent a starting point for future work.

- RC: Additionally, there is discussion of rapid rounding of granite pebbles within 8km of the source. I am interested whether the authors considered utilizing shorter distances, rather than 50km for the distance over which to fit a linear regression. In Figure 5a., the median roundness values appear to show the expected relationship where roundness increases toward 1 over the surveyed distance. Additionally, other studies (Miller, et al., 2014; Novak-Szabo, et al., 2018) observe the expected rounding curve over distances of 10km in the field. Since some granite pebbles are already fairly rounded at 8km in your study area, I expect that even this short distance is sufficient for noticeable shape changes to occur. Would the results change if the same analysis was applied over shorter distances? Would the results agree if the authors compared them to a more traditional rounding curve applied to the 50km over which the granite clasts were collected?
- AR: While we totally recognize that some pebbles may round over much shorted distances, we would like to highlight that the linear fit was used only to exemplify changes in the slope of the regressions, which motivated the development of our model. The model is non-linear (exponential) and as such, the best fit curve in Figure 8 (previously figure 7) is the one that best fits each percentile data with the exponential relationship we have chosen as the foundation for our model. This would be more obvious if we zoomed in small sections of the figure that contain only one set of percentile data. We don't think we have enough data points (in particular in the granite) to test whether the fit to one particular percentile is consistent with the roundness curve we have derived from the whole dataset.

**3.3. Technical Comments**

**RC: L14: In the abstract, you state that the roundness coefficient is 8x greater for granite pebbles, but later state that it is 7x greater.**

AR: After recalculation and check of all figures, we concluded that the ratio is nine (now corrected in all instances). We apologise for the inconsistency.

Our field data suggest that the roundness coefficient for granite pebbles is eight nine times that of quartzite pebbles.

**RC: L23: Recommend use of semicolon or em dash rather than colon as punctuation after "This also applies to modern rivers".**

AR: L23 in the original manuscript corresponds to line 22 in the revised manuscript.

This also applies to modern rivers: rivers where the shape of pebbles has been used to locate sediment sources and define the control exerted by hydraulic transport on abrasion processes

**RC:** *L26:* Same as above regarding the colon after "not limited to Earth".**

AR: L26 in the original manuscript corresponds to l

---

## Author Response (AR2)

**Authors' Response to Editors' comment of**

**Downstream rounding rate of pebbles in the Himalaya**

P. Pokhrel, M. Attal, H. D. Sinclair, S. M. Mudd and M. Naylor
*EGUsphere Preprint,* `https://doi.org/10.5194/egusphere-2023-2157`
* * *
**RC:** *Editors' Comment*,     AR: Authors' Response,     ☐ Manuscript Text

**1.  Editors' comment: Rebecca Hodge**

AR:  Dear Rebecca Hodge,

Thank you very much for handling our paper and for providing comments and suggestions that have contributed to enhancing the quality of our manuscript. We would also like to express our gratitude to entire editorial team for diligently conducting the review process efficiently. Your prompt and thorough feedback has been invaluable. Additionally, we appreciate the extension you provided, which allowed us the necessary time to incorporate the suggested improvements. Your support throughout the editorial process is truly appreciated.

Kind regards,
Pokhrel et al.

**RC:** *Public justification (visible to the public if the article is accepted and published):*

AR:  We agree to make it visible to the public if the article is accepted and published.

**RC:** *Thanks for your thorough and careful revision of your paper. Thanks also for providing a comprehensive response to reviewers document that makes it clear how you have addressed each comment. I have a handful of very minor edits to for you to consider, but otherwise I am happy to recommend this paper for publication and think that it will be a very useful addition to the research literature in this area.*

AR:  Thanks once again for providing comments and suggestions that have contributed to enhancing the clarity of our manuscript. We have provided a detailed response below.

**RC:** *Comments by line number (in the tracked changes version):*

**RC:** *53: You define roundness later on, but I think that it would be useful to add a brief definition here to clarify that you are talking about perimeter shape rather axes ratios.*

AR:

> There are different views regarding the controls on and trends in pebble roundness as one moves downstream . Figure 1 illustrates the downstream change in the perimeter shape of the pebbles.

**RC:** *100: Another reference that might be useful to add to this section is Bodek and Jerolmack (2021), who also used image analysis to quantify particle shape in rotating drum experiments and calculated IR: https://esurf.copernicus.org/articles/9/1531/2021/esurf-9-1531-2021.html*

AR:

> However, studies developing an automated workflow to reduce the subjectivity in calculating the shape parameters have been recently published  (e.g., (Roussillon et al., 2009; Cassel et al., 2018; Bodek and Jerolmack, 2021)). Roussillon et al. (2009) developed a tool for the automatic extraction of pebble shape from 2D images.

**RC:** *200: Do you mean perimeter instead of contour?*

**AR:** The definitions of curvature of contour and perimeter differ in Durian et al. (2006). They quantify the shape of flat pebbles by measuring curvature at each point along the entire two-dimensional contour. Curvature (K) is defined as the reciprocal radius of a circle that locally matches the contour, deduced from the coordinates of the pebble boundary (Weisstein, 1999). However, in our study, we consider the measurement of the pebble boundary itself as the perimeter. Due to this distinction, replacing the word "contour" with "perimeter" would not be appropriate. Therefore, we propose excluding the entire sentence from our manuscript to prevent any potential confusion.

>

**RC:** *337: Do you mean bounding instead of bonding?*

**AR:**

> Finally, the major ($a$) axis and intermediate ($b$) axis are measured using the """Minimum bounding geometry" function from the search box tool in ArcGIS, with "Geometry type" as convex hull and "Geometry characteristics" as attribute added.

**RC:** *414: I'm not sure what 'negative of the R-squared values' means, as I would assume that R-squared is always positive?*

**AR:** When using the downhill gradient method, such as the 'Nelder-Mead' method, the goal is to minimize a cost or objective function. In the context of maximizing R-squared, which is a measure of the goodness-of-fit in regression analysis, the convention is to frame the optimization problem as a minimization task. This is why the negative of the R-squared values is used in the optimization process. By minimizing the negative of the R-squared values, the optimization algorithm is effectively seeking parameter values that maximize the original R-squared value. This is a common approach in optimization problems, where the focus is on minimizing a cost function or, in this case, the negative of a performance metric.

**AR:** For clarity, the sentence now has been rephrased.

> This method aims to minimise the negative of the performance metric , effectively maximising the $R-squared$.

**RC:** *462: Change consist to consists.*

**AR:**

> The sample site is located in the Indo-Gangetic plain which consists of a full mixture of

sediments from Higher to Lesser Himalaya and Sub-Himalaya

**RC:**  *512: It's not clear to me what alluvial fans you are referring to. Maybe remove this?*

AR:

It is important to note that the granite clasts are absent in the alluvial fan deposits ~(top unit in Figure 12).~

**RC:**  *Figure 13: Use the same number of decimal places throughout the caption, so 0.975 rather than 0.98.*

AR:

Photograph showing the $IR_n$ value of the clasts at location "a" ($\sim$ 8 km downstream from channel head) in Figure 3 (d). Note that the roundness value for this location ranges from 0.867 to 0.975. Although the pebble with ~$IR_n = 0.98$~$IR_n = 0.975$ has travelled only 8 km from the channel head, its roundness is equivalent to that of pebbles which have travelled 50 km transport distance.

References:

Weisstein, E.W., 1999. CRC concise encyclopedia of mathematics. CRC press. In: Durian, D.J., Bideaud, H., Duringer, P., Schröder, A., Thalmann, F. and Marques, C.M., 2006. What is in a pebble shape?. Physical Review Letters, 97(2), p.028001.

Bodek, S. and Jerolmack, D.J., 2021. Breaking down chipping and fragmentation in sediment transport: the control of material strength. Earth Surface Dynamics, 9(6), pp.1531-1543.